# The RAM signaling pathway links morphology, thermotolerance, and CO$_2$ tolerance in the global fungal pathogen *Cryptococcus neoformans*

Benjamin J Chadwick[1], Tuyetnhu Pham[1], Xiaofeng Xie[2], Laura C Ristow[3], Damian J Krysan[3,4], Xiaorong Lin[1,2]*

[1]Department of Plant Biology, University of Georgia, Athens, United States; [2]Department of Microbiology, University of Georgia, Athens, United States; [3]Department of Pediatrics, Carver College of Medicine, University of Iowa, Iowa City, United States; [4]Department of Microbiology and Immunology, University of Iowa, Iowa City, United States

*For correspondence:
Xiaorong.Lin@uga.edu

**Abstract** The environmental pathogen *Cryptococcus neoformans* claims over 180,000 lives each year. Survival of this basidiomycete at host CO$_2$ concentrations has only recently been considered an important virulence trait. Through screening gene knockout libraries constructed in a CO$_2$-tolerant clinical strain, we found mutations leading to CO$_2$ sensitivity are enriched in pathways activated by heat stress, including calcineurin, Ras1-Cdc24, cell wall integrity, and *R*egulator of *A*ce2 and *M*orphogenesis (RAM). Overexpression of Cbk1, the conserved terminal kinase of the RAM pathway, partially restored defects of these mutants at host CO$_2$ or temperature levels. In ascomycetes such as *Saccharomyces cerevisiae* and *Candida albicans*, transcription factor Ace2 is an important target of Cbk1, activating genes responsible for cell separation. However, no Ace2 homolog or any downstream component of the RAM pathway has been identified in basidiomycetes. Through in vitro evolution and comparative genomics, we characterized mutations in suppressors of *cbk1Δ* in *C. neoformans* that partially rescued defects in CO$_2$ tolerance, thermotolerance, and morphology. One suppressor is the RNA translation repressor Ssd1, which is highly conserved in ascomycetes and basidiomycetes. The other is a novel ribonuclease domain-containing protein, here named *PSC1*, which is present in basidiomycetes and humans but surprisingly absent in most ascomycetes. Loss of Ssd1 in *cbk1Δ* partially restored cryptococcal ability to survive and amplify in the inhalation and intravenous murine models of cryptococcosis. Our discoveries highlight the overlapping regulation of CO$_2$ tolerance and thermotolerance, the essential role of the RAM pathway in cryptococcal adaptation to the host condition, and the potential importance of post-transcriptional control of virulence traits in this global pathogen.

## Editor's evaluation

This paper reports the identification of molecular determinants of CO2 tolerance in the human fungal pathogen Cryptococcus neoformans. The results are important for our understanding of how the fungus adapts from the ambient atmosphere to the CO2-enriched environment in the human host, and the findings are convincing and rely on biochemical, molecular, and genetic techniques. The results should be of interest to a broad community in the life sciences including microbiologists and infectious diseases investigators.

## Introduction

There are over 278,000 cases of cryptococcal meningitis every year, causing over 180,000 deaths (*Rajasingham et al., 2017*). Cryptococcal meningitis is primarily caused by the ubiquitous environmental fungus *Cryptococcus neoformans*. Airborne spores or desiccated yeast cells of *C. neoformans* are inhaled into the lungs, where they are cleared or remain dormant until reactivation upon host immunosuppression (*Casadevall and Perfect, 1998*; *Zhao et al., 2019*).

Litvintseva et al. found that most environmental *Cryptococcus* isolates cannot cause fatal disease in mouse models of cryptococcosis, despite having similar genotypes and in vitro phenotypes to known virulent isolates, including thermotolerance, melanization, and capsule production (*Litvintseva and Mitchell, 2009*). *Mukaremera et al., 2019* also observed that in vitro phenotype assays for thermotolerance, capsule production, titan cell formation, or fluconazole heteroresistance could not differentiate high-virulence strains from low-virulence strains. These observations raise the possibility that other, unidentified virulence traits are important for *Cryptococcus* pathogenesis. Tolerance to host levels of $CO_2$ (~5% $CO_2$ in the host vs. ~0.04% in ambient air) is likely a significant factor separating the potentially virulent natural isolates from the non-pathogenic environmental isolates that Litvintseva et al. tested (*Krysan et al., 2019*; *Litvintseva and Mitchell, 2009*).

The ability to adapt to host conditions is a prerequisite for cryptococcal pathogenesis. For instance, the ability of *C. neoformans* to replicate at human body temperature (≥37°C) has been extensively investigated. Many genes have been shown to be essential for thermotolerance (*Perfect, 2006*; *Stempinski et al., 2021*; *Yang et al., 2017*), including calcineurin which is currently being explored for antifungal drug development (*Gobeil et al., 2021*). By contrast, the underlying mechanisms or genes that play a role in $CO_2$ tolerance have yet to be identified. Here, we set out to identify $CO_2$-sensitive mutants and to gain the first insight into the genetic components involved in $CO_2$ tolerance in *C. neoformans*.

## Results

### $CO_2$ sensitivity is independent of pH

Our previous work indicates that many *C. neoformans* environmental strains are sensitive to 5% $CO_2$ when grown on buffered RPMI media, commonly used for mammalian cell cultures and testing antifungal susceptibility (*Krysan et al., 2019*). $CO_2$ at host concentrations also acts synergistically with

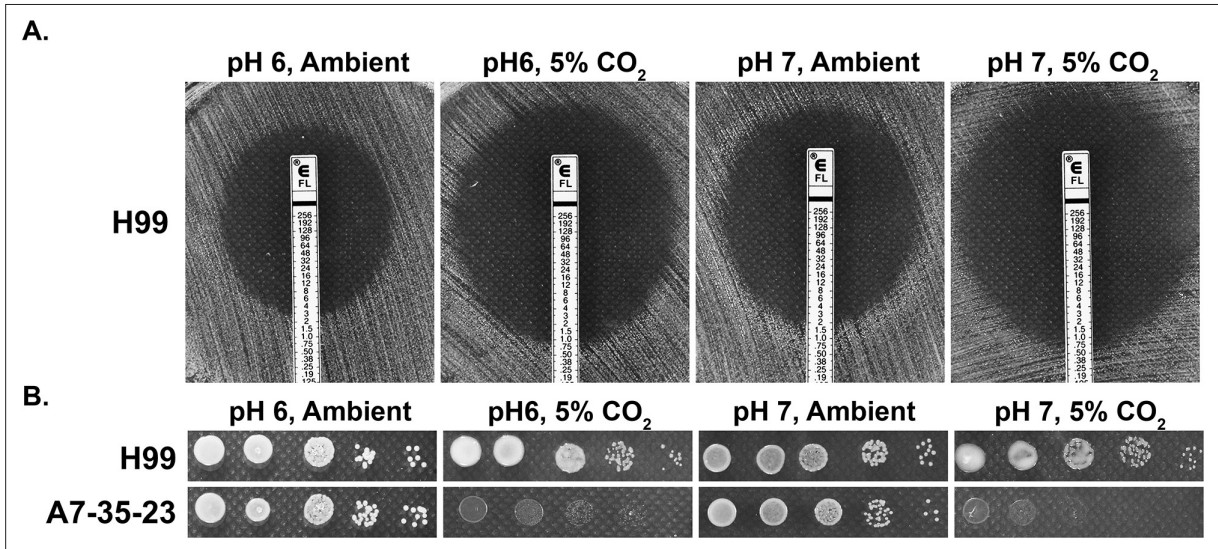

**Figure 1.** $CO_2$ sensitivity is not simply due to lowered medium pH. (**A**) H99 cells were plated onto RPMI solid medium buffered to either pH 6 or pH 7. Fluconazole containing E-test strips were placed onto the lawn of H99 cells, and the plates were incubated at 37°C in ambient air or in 5% $CO_2$. The larger the halo surrounding the E-strip, the more sensitive the cells are to fluconazole. The intercept value of the halo with the E-strip is the minimal inhibitory concentration. (**B**) Cells of the previously identified $CO_2$-tolerant H99 and $CO_2$-sensitive A7-35-23 were serial diluted, spotted onto RPMI media buffered to pH 6 or pH 7, and incubated at 37°C in ambient air or in 5% $CO_2$.

the commonly used antifungal drug fluconazole in inhibiting cryptococcal growth on buffered RPMI media. Because $CO_2$ lowers the pH of aqueous environments, it is possible that the $CO_2$ growth inhibitory effect or its synergy with fluconazole is simply due to lower medium pH. To address this question, we tested sensitivity to fluconazole of wild-type (WT) strain H99 using E-test on buffered RPMI media of either pH 6 or pH 7, with or without 5% $CO_2$. In this E-test, the size of halo (clearance zone) reflects fungal susceptibility to fluconazole. As shown in **Figure 1A**, clearance zones were much larger in 5% $CO_2$ relative to those in ambient air at both pH 6 and pH 7, indicating that $CO_2$ sensitizes cryptococcal susceptibility to fluconazole. Furthermore, $CO_2$ inhibits the growth of H99 at both pH 6 and pH 7 (smaller colony size in 5% $CO_2$ relative to that in ambient air). Additionally, growth of $CO_2$-sensitive environmental strain A7-35-23 (**Krysan et al., 2019**) was severely inhibited by 5% $CO_2$ at both pH 6 and pH 7 (**Figure 1B**). In general, *C. neoformans* grows better at acidic pH (can grow well

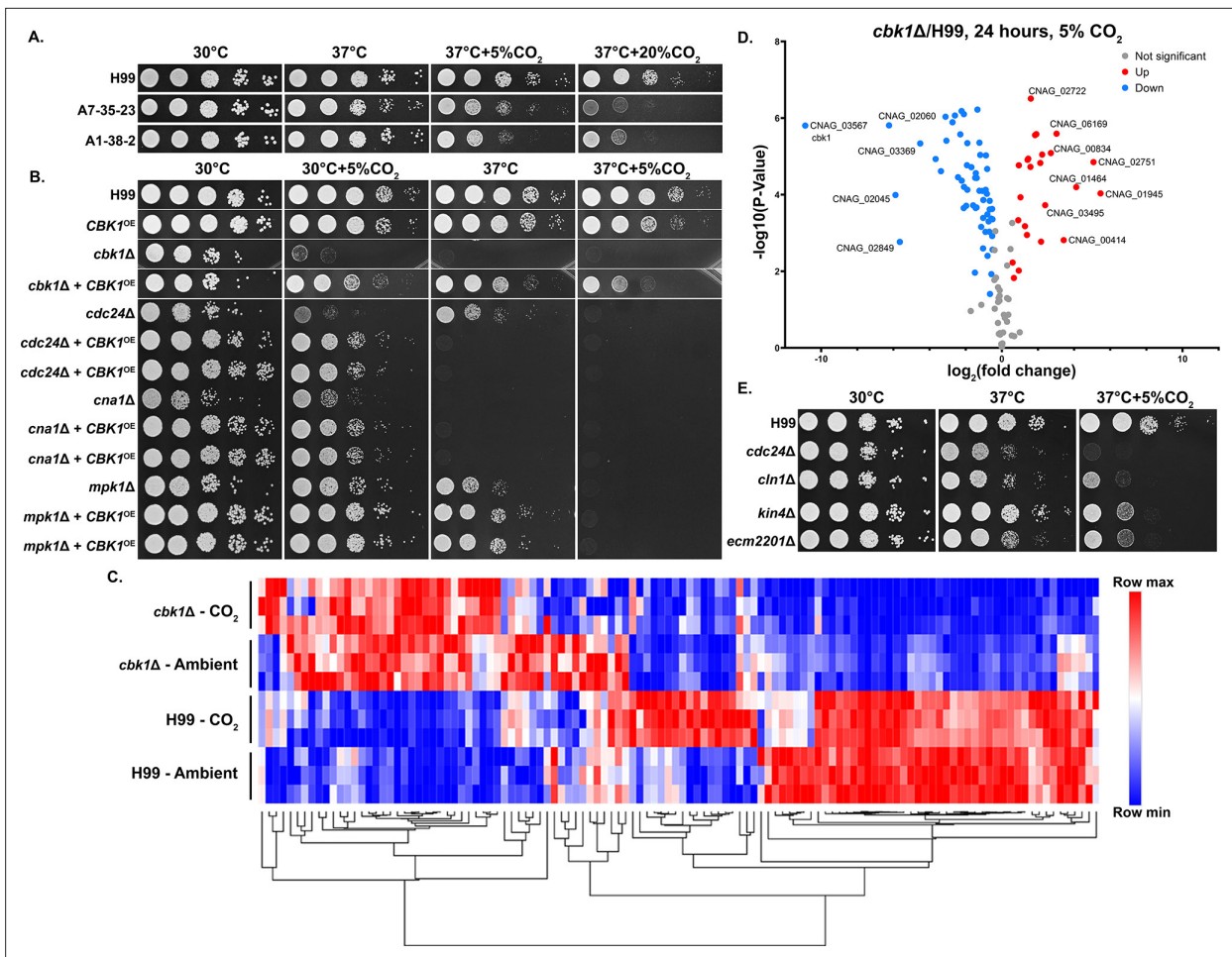

**Figure 2.** The *R*egulator of *A*ce2 and *M*orphogenesis (RAM) pathway effector kinase Cbk1 is critical for $CO_2$ tolerance. (**A**). The clinical reference strain H99 and environmental strains A7-35-23 and A1-38-2 were grown overnight in yeast peptone dextrose (YPD), serially diluted, and spotted onto solid YPD media plates. Photographs were taken 2 days after incubation in the indicated condition. (**B**) This serial dilution spotting assay was similarly performed for H99 and the mutants indicated. Two independent overexpression transformants for each mutant background were included as biological replicates. (**C**) Heatmap showing normalized total RNA counts of NanoString targets in H99 and *cbk1Δ* cultured at either ambient or 5% $CO_2$, red indicates higher and blue indicates lower transcript abundance. (**D**) Volcano plot showing significantly differentially expressed transcripts (p-value of <0.05) in the *cbk1Δ* compared to H99 in the 5% $CO_2$ condition. (**E**) Serial dilution spotting assay of H99 and four of the mutants found in the deletion set screening to be $CO_2$ sensitive which also correspond to significantly downregulated genes shown in the volcano plot.

The online version of this article includes the following figure supplement(s) for figure 2:

**Figure supplement 1.** The cAMP(cyclic AMP) pathway is not essential for $CO_2$ tolerance.

**Figure supplement 2.** Confirming overexpression of *CBK1*.

**Figure supplement 3.** Overexpression of *CDC24*, *MPK1*, or *CNA1* does not restore growth at host $CO_2$ or temperature levels.

in pH 3), and both A7-35-23 and H99 grew better at pH 6 than at pH 7 in ambient air (*Figure 1B*). Taken together, these results suggest that cryptococcal growth inhibition by $CO_2$ is not simply due to lowered pH.

## Identifying genes important for $CO_2$ tolerance

To identify genes involved in $CO_2$ tolerance in *C. neoformans*, we screened gene deletion mutants constructed in the $CO_2$-tolerant clinical reference strain H99. For large-scale screening, we used the nutrient rich yeast peptone dextrose (YPD) medium on which *C. neoformans* grows well. Accordingly, we tested the growth of two $CO_2$-sensitive environmental strains and the $CO_2$-tolerant H99 strain in different levels of $CO_2$ when cultured on YPD. As expected, relative to H99, the $CO_2$-sensitive strains A7-35-23 and A1-38-2 grew poorly at 5% $CO_2$ and worse at 20% $CO_2$ (*Figure 2A*). Using this approach, the following deletion mutant libraries were screened at 20% $CO_2$ on YPD media: a set of strains previously constructed in our lab, the collections constructed by the Madhani lab, and a set generated in the Lodge Lab (*Chun and Madhani, 2010*). As some mutants are known to be temperature sensitive, we carried out the screens at 30°C rather than 37°C. From over 5000 gene knockout mutants screened (~7000 protein coding genes in the H99 genome), 96 were found to be sensitive to $CO_2$ by visual observation (*Supplementary file 1*). We noticed that knockout mutants for multiple pathways known to be activated by heat stress are $CO_2$ sensitive, including the Ras1-Cdc24 pathway, calcineurin, cell wall integrity (CWI), and *R*egulator of *A*ce2 and *M*orphogenesis (RAM). This finding indicates an overlapping nature of these two traits.

We were surprised by the absence of components of adenylyl cyclase-PKA(protein kinase A) pathway from the set of hits. In *Candida albicans*, the adenylyl cyclase pathway is crucial for the yeast-hypha transition in response to host levels of $CO_2$ (*Klengel et al., 2005*). This pathway has also been proposed to play an important role for *Cryptococcus* to sense $CO_2$, and the carbonic anhydrase Can1 is required for growth at low concentrations of $CO_2$ (*Bahn et al., 2005*; *Mogensen et al., 2006*). However, we found that adenylyl cyclase pathway mutants showed no growth defects at host levels $CO_2$, including the adenylyl cyclase mutant *cac1Δ*, the adenylyl cyclase associated protein mutant *aca1Δ*, the alpha G protein subunit mutant *gpa1Δ*, and the cAMP-dependent protein kinase mutant *pkr1Δ* (*Figure 2—figure supplement 1*). This indicates that growth defects in response to host levels of $CO_2$ are likely independent of bicarbonate activation of adenylyl cyclase. This is not unexpected given that bicarbonate is not a limiting factor under the high level of $CO_2$ used in our screen.

Because the calcineurin, Ras1-Cdc24, CWI, and RAM pathways are all activated at host temperature and were identified in our screen for $CO_2$-sensitive mutants, we reasoned their downstream effectors may be related or genetically interact. As the RAM pathway effector kinase mutant *cbk1Δ* showed the most severe defect in thermotolerance and $CO_2$ tolerance compared to the mutants of the other pathways, we first overexpressed the gene *CBK1* in the following mutants, *cdc24Δ* (Ras1-Cdc24), *mpk1Δ* (CWI), *cna1Δ* (Calcineurin), and the *cbk1Δ* mutant itself and observed their growth at host temperature and host $CO_2$ (*Figure 2B*). Overexpression was achieved by placing the *CBK1* open reading frame after the inducible *CTR4* promoter, which is highly activated in YPD media (*Ory et al., 2004*; *Wang et al., 2014*; *Wang et al., 2012*). The *CBK1* overexpression construct was specifically integrated into the 'safe haven' locus *SH2* (*Lin et al., 2020*; *Upadhya et al., 2017*) in each mutant strain background to avoid complications due to positional effects. We additionally confirmed overexpression of *CBK1* by RT-PCR (*Figure 2—figure supplement 2*). As expected, the growth defects of the *cbk1Δ* mutant at 37°C with and without 5% $CO_2$ were largely restored by *CBK1* overexpression. At 30°C, overexpression of *CBK1* restored the growth of the *mpk1Δ* mutant, the *cna1Δ* mutant, and the *cdc24Δ* mutant in the $CO_2$ condition. In terms of thermotolerance, overexpression of *CBK1* restored growth of *mpk1Δ* but not *cna1Δ*, while the growth defect of *cdc24Δ* at 37°C was exacerbated. *CBK1* overexpression failed to rescue growth of any of these mutants when both stressors were present (37°C+5% $CO_2$). We found that overexpression of *CBK1* in the WT H99 background caused a modest growth defect at 37°C+5% $CO_2$. Thus, the detrimental effects from *CBK1* overexpression under this growth condition may partially explain its inability to fully rescue growth of these tested $CO_2$-sensitive mutants. The reciprocal overexpression of *CDC24*, *MPK1*, or *CNA1* in the *cbk1Δ* mutant background did not restore growth under 37°C and/or 5% $CO_2$ (*Figure 2—figure supplement 3*). These results support a hypothesis that Cbk1 integrates multiple stress response pathways to regulate both $CO_2$ tolerance and thermotolerance.

To determine the extent of Cbk1's role in $CO_2$ tolerance, we conducted NanoString gene expression profiling of the WT H99 and *cbk1Δ* mutant cultured in ambient air and in 5% $CO_2$ at 30°C (*Figure 2C*). Transcript levels of 118 genes were measured, and those genes were chosen based on RNA sequencing results from a separate study (Ristow et al., in preparation). In that study, these genes were differentially expressed in $CO_2$ vs. ambient air conditions in either two $CO_2$-sensitive or two $CO_2$-tolerant natural strains (*Source data 1*). Out of these 118 $CO_2$-associated genes, 81 were found to be significantly differentially expressed in the *cbk1Δ* mutant in both ambient air and in 5% $CO_2$, indicating they are intrinsically dysregulated in the *cbk1Δ* mutant. 57/81 of these genes are downregulated and 24/81 upregulated compared to the WT H99 strain (*Figure 2D*). Interestingly, 16/57 of the downregulated genes were also hits in our deletion set screening. We picked four of these deletion mutants which showed high sensitivity in our screen, to assay their sensitivity to host $CO_2$ conditions by spotting assay (*Figure 2E*). Taken together, this transcriptomic profiling shows that loss of Cbk1 significantly affects the expression of $CO_2$-related genes.

## The RAM signaling pathway is critical for normal morphology, thermotolerance, and $CO_2$ tolerance

The RAM pathway effector kinase Cbk1 is part of the NDR/LATS family of kinases, which is conserved from yeast to humans and affects a wide range of cellular functions including cell-cycle regulation. In *C. neoformans*, various virulence factors are impacted by deletion of *CBK1*, including urease activity and thermotolerance (*Lee et al., 2016*). Through our genetic screen for $CO_2$-sensitive mutants, we found that all tested *Cryptococcus* RAM pathway mutants are extremely sensitive to 5% $CO_2$ and high temperature, and they show no growth at 37°C+5% $CO_2$ (*Figure 3A*). In ascomycetes such as *Saccharomyces cerevisiae* and *C. albicans*, RAM pathway mutants are defective in cytokinesis and exhibit loss of polarity, resulting in enlarged round cells that cluster together (*Saputo et al., 2012*; *Figure 3—figure supplement 1A*). In contrast, though defective in cytokinesis (*Magditch et al., 2012*; *Walton et al., 2006*), *Cryptococcus* RAM pathway mutants are hyper-polarized and constitutively form clusters of elongated pseudohyphal cells (*Figure 3B*). Moreover, we found that while the *C. albicans* homozygous *cbk1ΔΔ* mutant exhibits a general growth defect compared to the WT control, it shows no apparent specific growth defect at 37°C with or without 5% $CO_2$ (*Figure 3—figure supplement 1C*). These results suggest that, although the RAM pathway is conserved in its role in cytokinesis, the effects of its downstream targets are divergent between ascomycetes and basidiomycetes.

## Suppressors of the *cbk1Δ* mutant show improved growth at host conditions

In ascomycetes, Ace2 is a key downstream transcription factor of the RAM pathway (hence in the name of RAM − *R*egulator of *A*ce2 and *M*orphogenesis), which is important for the activation of genes responsible for cell separation as well as a large number of genes with other functions (*Mulhern et al., 2006*; *Wakade et al., 2020*). However, no homolog to Ace2 has been identified in *Cryptococcus* or other basidiomycetes. Furthermore, no downstream targets of the RAM pathway have been identified in any basidiomycetes. To investigate potential downstream effectors of the RAM pathway in *Cryptococcus*, we screened for spontaneous suppressor mutants of *cbk1Δ*. To do so, *cbk1Δ* mutant cells from an overnight culture in liquid YPD at 30°C were plated onto solid YPD media and incubated for 2 days at 37°C+5% $CO_2$. Out of >1×$10^8$ cells plated and cultured under this condition that is inhibitory for growth of the original *cbk1Δ* mutant, 11 suppressor colonies were isolated for further examination and sequencing. All the suppressor isolates showed dramatically improved growth over the original *cbk1Δ* mutant at 37°C and modestly improved growth at 37°C+5% $CO_2$ (*Figure 4C*). Based on their distinctive phenotypes, the 11 suppressors were classified into two groups: *sup1* (2/11) and *sup2* (9/11). Shorter chains of cells in both groups indicate a partial restoration in cytokinesis (*Figure 4D*). The *sup2* group has slightly improved growth at 37°C+5% $CO_2$ and forms shorter chains of cells compared to the *sup1* group (*Figure 4C and D*). Besides of these observations, *sup1* and *sup2* displayed similar phenotypes in growth assays including the cryptococcal virulence traits tested, including melanin production, capsule, urease activity, and cell wall stress tolerance. (*Figure 4—figure supplement 1*). Both *sup1* and *sup2* showed no improved growth compared to the *cbk1Δ* mutant at pH 7.4 37°C+5% $CO_2$. This is likely due to the detrimental combination of high temperature, $CO_2$, and high pH, as the WT also showed significantly reduced growth in this condition. (*Figure 4—figure supplement 1A*).

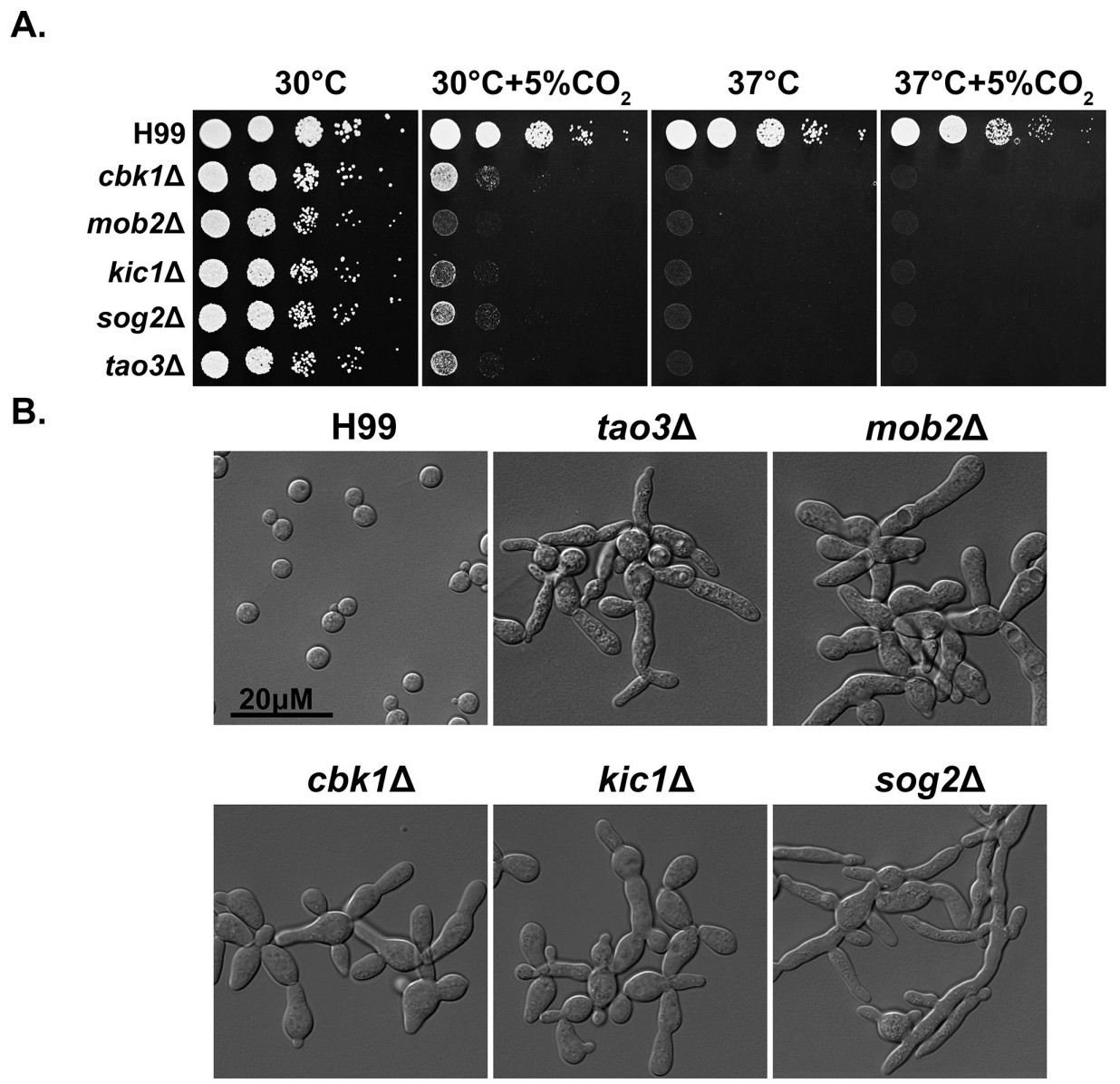

**Figure 3.** The *R*egulator of *A*ce2 and *M*orphogenesis (RAM) pathway is critical for normal morphology, thermotolerance, and $CO_2$ tolerance. (**A**) *Cryptococcus neoformans* WT H99 and RAM pathway mutants were serially diluted, spotted onto yeast peptone dextrose (YPD) medium, and incubated for 2 days at the indicated condition. (**B**) The cellular morphology of *C. neoformans* WT H99 and RAM pathway mutants cultured in YPD medium.

The online version of this article includes the following figure supplement(s) for figure 3:

**Figure supplement 1.** Conserved and divergent roles of the *R*egulator of *A*ce2 and *M*orphogenesis (RAM) pathway in ascomycete *Candida albicans* and basidiomycete *Cryptococcus neoformans*.

Because RAM pathway suppressor mutants were previously identified after treatment with calcineurin inhibitor FK506 and showed improved growth in FK506 and restored mating (***Magditch et al., 2012***), we also tested our suppressors' growth in FK506 and their ability to mate. We found that both *sup1* and *sup2* failed to restore growth of the *cbk1Δ* on media supplemented with FK506 or restore the ability to mate with the congenic strain H99a (***Figure 4—figure supplement 1***).

Along with the original *cbk1Δ* mutant, we sequenced the genomes of the 11 *cbk1Δ* suppressors. By comparing their genome sequences with each other and with the original *cbk1Δ* mutant, we found that both *sup1* type suppressor mutants contained a disruptive in-frame deletion at the same location

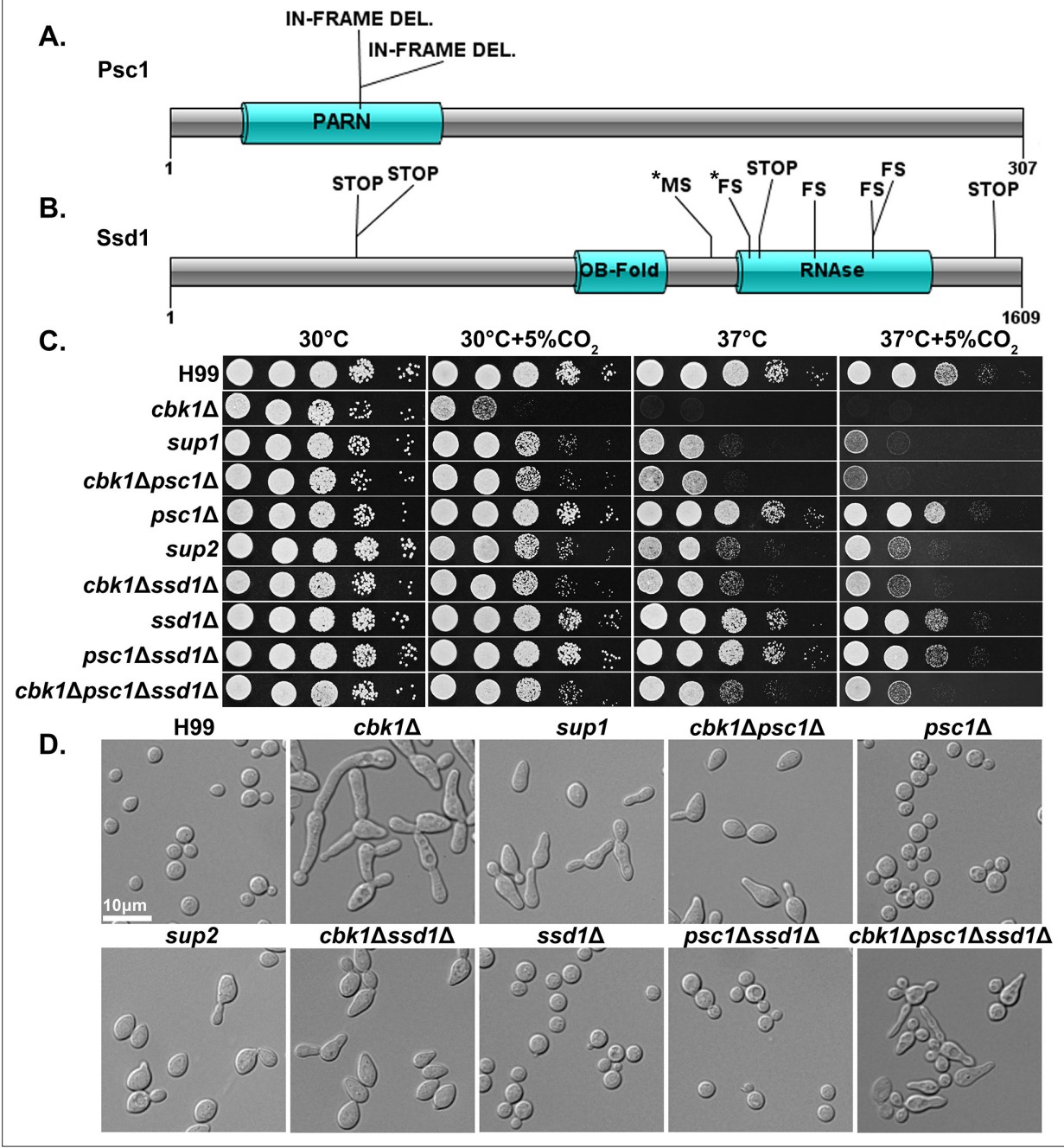

**Figure 4.** Natural suppressors of the *R*egulator of *A*ce2 and *M*orphogenesis (RAM) pathway *cbk1Δ* mutant restore multiple defects. (**A**) Protein diagram of Psc1 showing the effects and positions of suppressor mutations in the two *sup1* type natural suppressors. (**B**) Protein diagram of Ssd1 and the effects and positions of suppressor mutations in Ssd1 in the nine *sup2* type natural suppressors. STOP indicates a non-sense mutation, MS a missense mutation, and FS a frameshift mutation. Asterisks (*) indicate mutations in the same suppressor strain. (**C**) Serial dilutions of H99 and the mutant strains were spotted onto yeast peptone dextrose (YPD) agar media and incubated for 2 days in the indicated condition to observe growth. (**D**) The cellular morphology of H99 and the mutant strains in liquid YPD cultures were examined under microscope.

The online version of this article includes the following figure supplement(s) for figure 4:

**Figure supplement 1.** Phenotypic characterization of *cbk1Δ* suppressor mutants.

**Figure supplement 2.** Suppressor mutants do not restore transcript levels of NanoString targets in *cbk1Δ*.

in *CNAG_01919*, which encodes a putative Poly(A)-specific ribonuclease (PARN) domain-containing protein (*Figure 4A*). This domain was previously reported in *S. pombe* proteins (*Marasovic et al., 2013*). Interestingly, through a BLAST search of the PARN domain, we did not identify this domain in any protein in the genomes of *S. cerevisiae*, *C. albicans*, or other ascomycetes but found it in basidiomycetes and higher eukaryotes. The in-frame deletion results in a change of two amino acids within the predicted PARN domain, the only discernable domain present in this protein. We named this previously uncharacterized gene *Partial Suppressor of cbk1* (*PSC1*). All nine *sup2* isolates contained loss of function or missense mutations in the gene *CNAG_03345* (*Figure 4B*), which encodes an RNA-binding protein homologous to *S. cerevisiae* Ssd1p, a known suppressor of *cbk1Δ* phenotypes in *S. cerevisiae*. ScSsd1p represses transcript translation and is negatively regulated by Cbk1p phosphorylation (*Jansen et al., 2009*; *Wanless et al., 2014*).

To confirm that the putative loss-of-function mutations in *SSD1* and *PSC1* are responsible for suppressing *cbk1Δ* phenotypes, we created *cbk1Δssd1Δ* and *cbk1Δpsc1Δ* double mutants together with the control single mutants *ssd1Δ* and *psc1Δ*. Indeed, relative to the *cbk1Δ* mutant, the double mutants showed reduced sensitivity to host temperature and $CO_2$ levels (*Figure 4C*), similar to the natural suppressor mutants. Likewise, the morphology of the double mutants resembles that of the spontaneous suppressor mutants (*Figure 4D*). The deletion of *SSD1* and *PSC1* alone in the WT background did not yield any discernable phenotype. The results confirm that loss-of-function mutations in *SSD1* and *PSC1* are responsible for partial suppression of the *cbk1Δ* mutant's growth defects observed in the isolated suppressor strains. Interestingly, *sup2* and the *cbk1Δssd1Δ* mutants both grew noticeably better than *sup1* and *cbk1Δpsc1Δ* at 37°C and 37°C+5% $CO_2$. To test the genetic interaction between the two suppressor genes *SSD1* and *PSC1*, we created a triple *cbk1Δpsc1Δssd1Δ* mutant and the control strain *psc1Δssd1Δ*. The *psc1Δssd1Δ* control strain did not exhibit any defect and grew similarly well to either single mutant or the WT (*Figure 4C*). The triple mutant *cbk1Δpsc1Δssd1Δ* grew similarly well as *sup2* or *cbk1Δssd1Δ* at 37°C+5% $CO_2$ (*Figure 4C*). However, the triple mutant displayed aberrant morphology and budding defects which are not observed in the natural suppressor mutants or the *cbk1Δssd1Δ* and *cbk1Δpsc1Δ* double mutants (*Figure 4D*). These results suggest that Psc1 and Ssd1 may function in the same pathway in regulating thermotolerance and $CO_2$ tolerance, but their downstream effects on cell separation and/or polarized growth may be overlapping and distinct.

To determine if the suppressor mutations restore transcript abundance of the differentially expressed genes under $CO_2$ in *cbk1Δ*, we compared the profiles of *cbk1Δ* to the two suppressor mutants: *sup1* and *sup2*. Overall, we found that the spontaneous suppressors do not restore transcript abundances of most differentially expressed genes in *cbk1Δ* to WT levels (*Figure 4—figure supplement 2*), suggesting that suppressors affect post-transcriptional regulation of $CO_2$ tolerance.

## Spontaneous suppressors of *cbk1Δ* mutant show improved ability to survive and replicate in the host

RAM mutants have previously been found to be attenuated in virulence in the invertebrate wax moth larva infection model and mouse intranasal infection models (*Lee et al., 2016*; *Magditch et al., 2012*). Occasionally, cryptococcal strains with point mutations in RAM genes cause death of mice when revertant mutations occur, which restore the function of the RAM pathway (*Magditch et al., 2012*). As shown above and consistent with previous literature, the *cbk1Δ* mutant shows a severe growth defect at host temperature and $CO_2$ concentrations (*Figure 3C*). Because *sup1* and *sup2* both largely restored growth to the cbk1Δ mutant at 37°C but only modestly restored growth at 37°C+5% $CO_2$, we decided to test if, and by how much, these suppressor mutations would affect the virulence of the *cbk1Δ* mutant. We infected mice with $1×10^4$ cells of WT, *cbk1Δ*, *sup1*, or *sup2* intranasally. In this intranasal infection model, the WT H99 strain establishes lung infection first and typically disseminates to other organs including the brain by 7–10 days post-infection (DPI). Mice infected by H99 normally become morbidly ill by 3–4 weeks post-infection and have a high fungal burden in the lungs, brain, and kidney (*Chadwick and Lin, 2020*; *Lin et al., 2022*).

As expected, all mice infected with H99 were moribund by DPI 26 (*Figure 5A*), while those infected with the *cbk1Δ* mutant survived until the experiment was terminated at DPI 60. Surprisingly, *sup1* and *sup2* strains did not cause any mortality either. The organ fungal burden, however, revealed differences in virulence levels between these strains. At the time of euthanasia for H99-infected mice

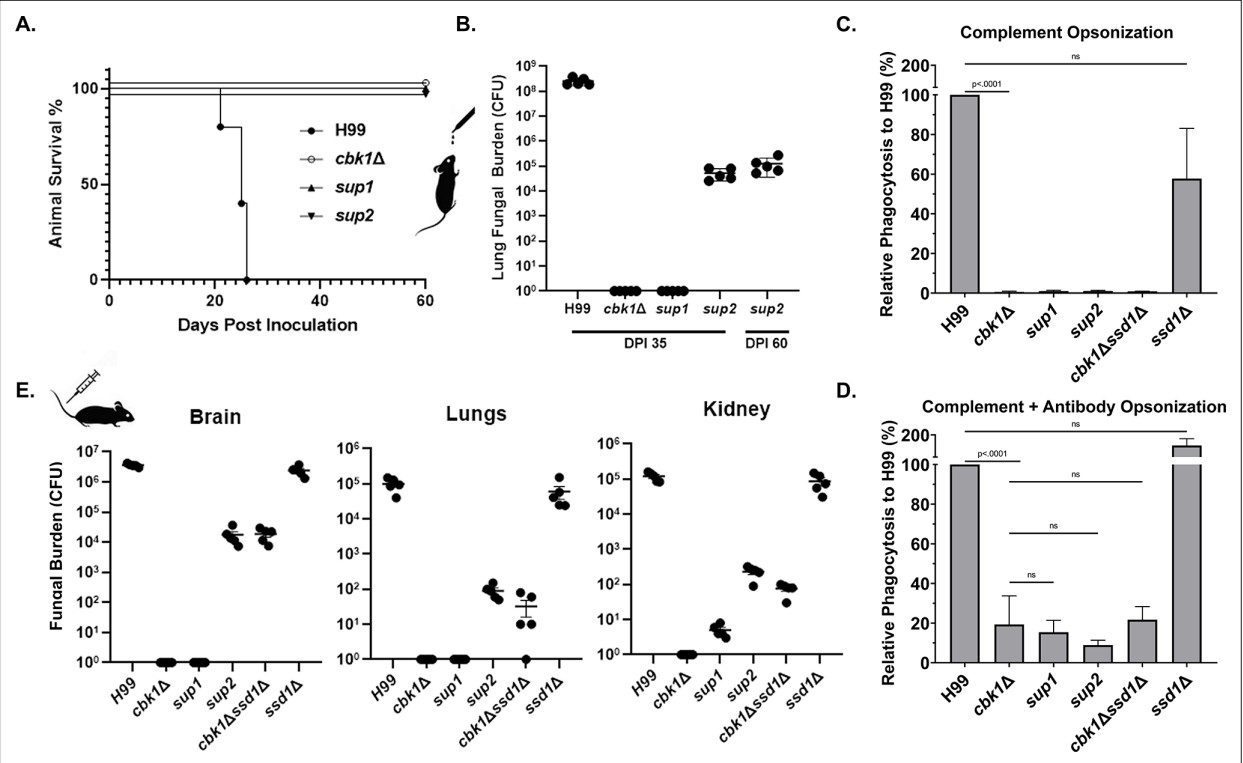

**Figure 5.** Suppressor mutants are partially restored for phagocytosis and can disseminate in the intravenous infection model of cryptococcosis. (**A**) Mice were infected with $1\times10^4$ fungal cells intranasally, and their survival was monitored for 60 days post-infection. (**B**) At day 35 post-infection (DPI 35) and at the time of termination (DPI 60), 5 out of 10 mice per group for the *cbk1Δ* mutant, *sup1* and *sup2* groups were harvested for brains, kidneys, and lungs. For H99 infected mice, they were euthanized at their clinical end point (all before DPI 26). Tissue homogenate was serially diluted and plated onto YNB(yeast nitrogen base) medium to count the colony-forming units (CFUs) to measure the fungal burden per organ. (**C**) Murine macrophage J774A.1 cells were co-incubated with $2\times10^6$ cryptococcal cells opsonized with serum from naïve mouse for 2 hr. Non-adherent or phagocytosed cells were washed, and cryptococcal cells were released and then serially diluted before plating onto YNB medium for measurement of CFUs. (**D**) The same as above, except opsonization, was performed with serum of mice vaccinated against cryptococcosis. (**E**) Mice were challenged with $1\times10^5$ cryptococcal cells intravenously. At day 5 post-infection, five mice per group were sacrificed. Brains, kidneys, and lungs of euthanized mice were dissected and homogenized. Serial dilutions were plated to count CFUs for quantification of fungal burden per organ.

(prior to DPI 26), the median fungal burden in the lungs, brains, and kidneys was $2.1\times10^8$, $1.4\times10^6$, and $2.4\times10^4$ colony-forming units (CFUs) per organ, respectively (*Figure 5B*). As expected, mice completely cleared the *cbk1Δ* mutant at DPI 35. Surprisingly, despite largely restored growth at 37°C, *sup1* was completely cleared from the mouse lungs by DPI 35, similar to the *cbk1Δ* mutant. In comparison, although *sup2* did not cause any death during the study period, it was able to replicate in the mouse lungs. The median lung fungal burden at DPI 35 was $8.2\times10^4$, over eightfold higher than the original inoculum. The *sup2* strain maintained the same high lung fungal burden at DPI 60 (*Figure 5B*), indicating that it can persist in the lung tissue. The only in vitro difference observed between *sup1* and *sup2* was better growth of *sup2* at host $CO_2$ levels which may explain the difference in their ability to propagate and persist in the mouse lung. However, it is worth nothing that due to the complex host environment, there could be other unrecognized factors contributing to the differences in vivo.

Although the spontaneous suppressor *sup2* was able to replicate in the mouse lungs, no fungal burden was detected in the brain or the kidney at DPI 35 or 60 (no organisms were detected in any of the mice), indicating that the mutant was unable to disseminate. We considered two hypotheses: (1) inability of suppressor *sup2* to disseminate from the lungs; (2) inability of suppressor *sup2* to penetrate other organs from the blood. Because *C. neoformans* can disseminate from the lungs to other organs by a 'Trojan Horse' mechanism, where *Cryptococcus* travels within the mobile host phagocytes (*Kechichian et al., 2007*; *Santiago-Tirado et al., 2017*), we examined phagocytosis of the *cbk1Δ* mutant and its suppressors to test the first hypothesis. We expected that cryptococcal mutants defective in being phagocytosed by host cells might be defective in dissemination, and the *cbk1Δ* mutant

was previously found to have a poor phagocytosis index (*Lin et al., 2015*). Here, we co-cultured murine macrophage JA774 cells with H99, *cbk1Δ*, *sup1*, *sup2*, the double mutant *cbk1Δssd1Δ*, or the control single mutant *ssd1Δ*. Because different types of opsonization can impact phagocytosis of *C. neoformans*, opsonization was performed using either naïve mouse serum (complement mediated phagocytosis) or serum from mice vaccinated against cryptococcosis (complement+antibody mediated phagocytosis). The serum (containing antibodies) from the vaccinated mice recognizes antigens present in the capsule of cryptococcal cells (*Lin et al., 2022*; *Zhai et al., 2015*). Consistent with our previous finding, phagocytosis of the *cbk1Δ* mutant was extremely low (~1% of the WT H99 level under complement mediated phagocytosis, *Figure 5C*). Opsonization with serum from vaccinated mice increased phagocytosis of *cbk1Δ* and the suppressor mutants, but the phagocytosis indexes of these mutants were still only 20% or less than that of the WT (*Figure 5D*). In both phagocytosis experiments, the suppressor mutants or the double mutants *cbk1Δssd1Δ* and *cbk1Δpsc1Δ* mutants showed increased phagocytosis relative to the *cbk1Δ* mutant. The poor phagocytosis of the *cbk1Δ* mutant and its suppressors may contribute to their lack of dissemination from the lungs to the other organs in the inhalation infection mouse model of cryptococcosis.

To test the second hypothesis, we infected mice intravenously with H99, *cbk1Δ*, *sup1*, *sup2*, the double mutant *cbk1Δssd1Δ*, or the control single mutant *ssd1Δ*. In this intravenous infection model, the barrier of the lungs is bypassed. H99 cells disseminate to the brain and other organs within hours (*O'Connor et al., 2013*). Because H99 rapidly disseminates in this model, infected mice typically reach moribundity after 1 week. Therefore, we euthanized mice at DPI 5 before H99-infected mice would have become moribund. As expected, H99-infected mice showed high fungal burdens in the lungs, brains, and kidneys, with the highest fungal burden in the brain (over $10^6$ CFUs; *Figure 5E*). The *cbk1Δ* mutant failed to disseminate in this intravenous infection model as no viable cells were recovered in any organ. Similarly, we could not recover any *sup1* cells from the lungs or the brain and only detected a few fungal cells in the kidney. In contrast, *sup2* suppressor mutants were recovered in all three organs, albeit with reduced fungal burdens (~$10^4$ CFUs in the brain and a few hundred in lungs/kidney) compared to the WT H99 control group (*Figure 5E*). This finding indicates that the *sup2* suppressor, once disseminated into the bloodstream, can invade other organs and replicate. Combined with the earlier observations that (1) both suppressors fully restore growth at host temperature and (2) *sup2* is slightly more $CO_2$ tolerant than *sup1*, the observation that only *sup2* can survive, amplify, and persist in animals implicates an importance of $CO_2$ tolerance in cryptococcal pathogenesis. Collectively, the

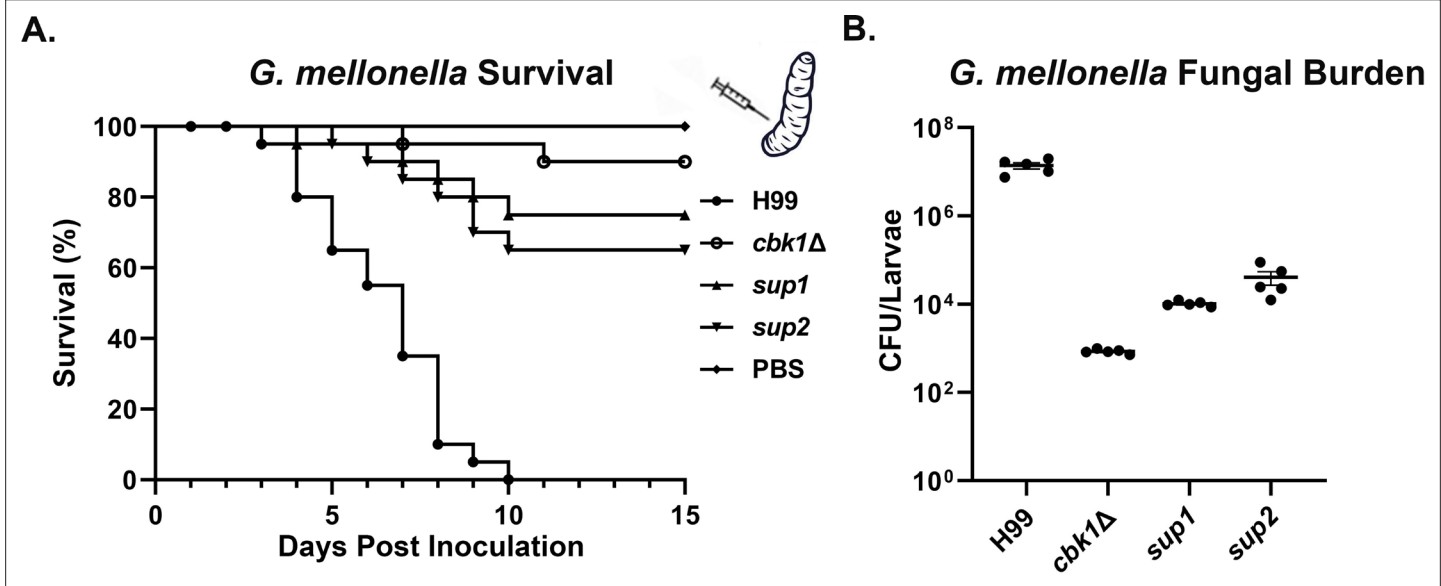

**Figure 6.** Suppressor mutants partially restore virulence in the *G. mellonella* model. (**A**) *G. mellonella* larvae were infected with 5×$10^4$ fungal cells of the indicated strain, and their survival was monitored for 15 days post-inoculation. The *cbk1Δ* group survival curve was not significantly different from *sup1* (p-value=0.2) and was significantly different from the *sup2* curve (p-value=0.05). (**B**) At day 5 post-inoculation (DPI 5), 5 out of 25 larvae per group for the H99, the *cbk1Δ* mutant, *sup1*, and *sup2* groups were homogenized, serially diluted, and plated to count the colony-forming units (CFUs) and measure the fungal burden of each larva. The fungal burden of the *sup1* and *sup2* groups was significantly higher than the *cbk1Δ* group (p-value<0.001).

results from phagocytosis, the inhalation infection model, and the intravenous infection model support the hypothesis that failure of the suppressor mutants to disseminate to other organs in the intranasal model is largely due to reduced phagocytosis and inability to escape the lungs. That said, other factors, such as increased systemic clearance by the immune system, could potentially contribute to the containment of the mutant in the lungs. Again, the cbk1Δssd1Δ mutant recapitulated the pheno-type of the sup2 strain in intravenous infection model and other in vitro assays, demonstrating that our observed sup2 phenotypes are due to disruption of SSD1.

As mutants that are temperature sensitive have reduced virulence in the mouse model of crypto-coccosis, we decided to test the virulence of these strains in the *Galleria mellonella* larvae infection model to remove temperature as a variable. We inoculated *G. mellonella* larvae with $5 \times 10^4$ cells of WT, cbk1Δ, sup1, or sup2 and maintained the larvae at 30°C as previously described (*Magditch et al., 2012*). PBS buffer inoculated larvae were included as a sham control. Infected larvae (n=20 per strain) were monitored for survival over a period of 15 days post-inoculation. All 20 of the larvae inoculated with the H99 strain died between DPI 3 and DPI 10 (*Figure 6A*). In comparison, only 2/20 of the cbk1Δ mutant-infected larvae died during this period. Interestingly, 5/20 sup1-infected larvae and 7/20 sup2-infected larvae died in this experiment (*Figure 6A*), indicating their partially restored viru-lence in this larva infection model. To further confirm the observed differences between these strains in this model conducted at 30°C, we infected five larvae with $5 \times 10^4$ cells per strain and measured their fungal burden at day 5 post-inoculation. At DPI 5, the mean fungal burden of WT-infected larvae was $1.5 \times 10^7$ CFUs (*Figure 6B*). In comparison, the mean fungal burden for the cbk1Δ-infected larvae was only $8.4 \times 10^2$ CFUs, which is almost 20,000-fold lower than the WT control group and about 60-fold lower than the original inoculum, indicating that most cbk1Δ cells have been cleared by this time point. The mean fungal burden of sup1-infected larvae was $9.6 \times 10^3$ CFUs, while the mean fungal burden sup2-infected larvae was $2.3 \times 10^4$ CFUs (*Figure 6B*). These results indicate that both sup1 and sup2 partially rescued virulence of the cbk1Δ mutant and that sup2 showed slightly better restoration of virulence compared to sup1 in this insect model, which is independent of tolerance to mammalian body temperature.

## Discussion

Detection of and adaptation to changing $CO_2$ levels are an important trait across biological kingdoms and may play a crucial role in the pathogenicity of fungi (*Bahn and Mühlschlegel, 2006*; *Cummins et al., 2014*; *Hetherington and Raven, 2005*; *Krysan et al., 2019*). Here, we report the identification of genes required for growth at high levels of $CO_2$ in the fungal pathogen *C. neoformans*. Multiple pathways important for growth at high temperature, such as the Ras1-Cdc24, CWI, Calcineurin, and RAM pathways, were found to be required for growth in high $CO_2$ concentrations, indicating that growth in response to host $CO_2$ may be intricately coordinated and co-regulated with response to host temperature. It is therefore likely that both host $CO_2$ and host temperature represent stressors that cryptococcal cells infecting mammalian hosts must overcome to cause disease.

Calcineurin and RAM pathways were both identified in our screen for mutants that affect crypto-coccal $CO_2$ sensitivity. A previous study found synthetic lethality between the RAM and calcineurin pathways in *C. neoformans* but not in *S. cerevisiae* (*Walton et al., 2006*). This corroborates our find-ings of the key differences between the basidiomycete *C. neoformans* and the ascomycete yeasts. In *C. albicans*, $CO_2$ levels are sensed through bicarbonate or cAMP-dependent activation of adenylyl cyclase to increase hyphal growth (*Du et al., 2012*; *Hall et al., 2010*). While these pathways may also be functioning to sense $CO_2$ in *Cryptococcus* (*Bahn et al., 2005*; *Mogensen et al., 2006*), our results indicate that these pathways do not play a significant role in host $CO_2$ tolerance in *C. neoformans*. We also found that disruption of the RAM pathway effector kinase Cbk1 caused a severe growth defect at host $CO_2$ in *C. neoformans* but not in *C. albicans*. The vast differences between these organisms in terms of growth response to $CO_2$ may reflect the evolutionary distance between these species and/or the distinct niches they normally occupy. Indeed, *C. albicans* is a human commensal and has adapted to host $CO_2$ concentrations. *S. cerevisiae* is a powerful fermenter that thrives in conditions with high levels of $CO_2$. For the environmental fungus *C. neoformans*, however, the ability to grow in a $CO_2$-enriched condition does not appear to be strongly selected for in the natural environment, and the host level of $CO_2$ (~5% $CO_2$) is over 100-fold higher than the ambient air (~0.04% $CO_2$).

The RAM pathway mutants were among the most sensitive mutants to host levels of $CO_2$. Remarkably, the growth defects of cbk1Δ could be partially restored by single mutations in the genes *PSC1* or *SSD1*. While the PARN-encoding gene *PSC1* represents an uncharacterized protein, *SSD1* is a known suppressor of cbk1Δ phenotypes that has been extensively characterized in ascomycete yeasts to regulate the translation of numerous and diverse mRNA transcripts (*Hu et al., 2018*; *Jansen et al., 2009*; *Lee et al., 2015*; *Li et al., 2009b*; *Wanless et al., 2014*). Our genetic interaction analysis indicates that Psc1 likely functions in the same pathway as Ssd1. Interestingly, in *S. cerevisiae*, deletion of *SSD1* can suppress the lethality of the cbk1Δ mutant but not the cell separation defect, which is regulated by the transcription factor Ace2 (*Kurischko et al., 2005*). However, an Ace2 homolog has not been identified in *Cryptococcus* or any other basidiomycete (*Lin et al., 2015*). In *C. albicans*, Ssd1 plays an important role in polarized growth and hyphal initiation by negatively regulating the transcription factor Nrg1 (*Lee et al., 2015*). The observation that cbk1Δpsc1Δ and cbk1Δssd1Δ suppressor mutants partially rescue cell separation defects or depolarized growth suggests that *C. neoformans* may primarily utilize Ssd1/Psc1 rather than a potential Ace2 homolog to regulate cell separation or polarization. Differential regulation of target mRNA transcripts by Ssd1 and Psc1 may explain the functional divergence of the RAM pathway we observed between the basidiomycete *Cryptococcus* and the ascomycete yeasts. Our observation that the natural suppressors do not restore transcript abundances of $CO_2$-associated genes in cbk1Δ to WT levels supports a hypothesis that disruption of Ssd1 and Psc1 suppresses the cbk1Δ mutant's defects at a post-transcriptional level. *C. neoformans* has been demonstrated to use post-transcriptional regulation to adapt to various host stresses (*Bloom et al., 2019*; *Kalem et al., 2021*; *Stovall et al., 2021*). A temperature-sensitive environmental species of *Cryptococcus*, *Cryptococcus amylolentus*, fails to initiate host stress-induced translational reprogramming and is non-pathogenic (*Bloom et al., 2019*). Whether or not translatome reprogramming is initiated in *C. neoformans* in response to host $CO_2$, and whether such reprogramming, if occurs, relies on Ssd1 and/or Psc1, has yet to be determined.

# Materials and methods

## Strains, growth conditions, and microscopy examination

Strains used in this study are listed in the key resources table. Unless stated otherwise, all *C. neoformans* cells were maintained at 30°C on YPD media or YPD + $CuSO_4$ (25 μM) for strains transformed with $P_{CTR4}$-*CBK1*. For morphological examination, all strains were examined under a Zeiss Imager M2 microscope, equipped with an AxioCam MRm camera. For spotting assays, the tested strains were grown overnight in liquid YPD medium at 30°C with shaking at 220 RPM. The cells were then adjusted to the same cell density of $OD_{600}$=1 and serially diluted 10-fold. The cell suspensions were then spotted onto YPD agar medium and incubated at the indicated condition for 2 days. $CO_2$ levels were controlled by a VWR $CO_2$ incubator or by a Pro-$CO_2$ controller (Biospherix, Lacona, NY, USA).

## Genetic manipulation

### Gene deletion constructs

To delete the gene *SSD1*, a deletion construct with a nourseothricin (NAT) resistance marker cassette with 5' and 3' homology arms to *SSD1* was used. Primers Linlab7974 (gctgcctttgcgtcatctc) and Linlab7976 (ctggccgtcgttttactctcgccttcttctcctta) were used to amplify the 5' arm from the H99 genome. The 3' arm was amplified from H99 with primers Linlab7977 (gtcatagctgtttcctgcgattgacatt gccgtcttag) and Linlab7979 (cgacctgatcaaactactcgc). The NAT marker was amplified with universal primers M13F and M13R from plasmid pPZP-NATcc. The three pieces were fused together by overlap PCR and amplified with nested primers Linlab7975 (acaatgagccactgccag) and Linlab7977 (tgcgtgtt cactactgtagac). To disrupt the gene *PSC1*, a hygromycin (HYG) marker cassette was used to insert into the PARN domain. To generate the sgRNA for specific targeting to the *SSD1* locus, the *U6* promoter and sgRNA scaffold were amplified from JEC21 genomic DNA and the plasmid pDD162 using primers Linlab7980/Linlab4627 (ttgagtggggtgggtcaattaacagtataccctgccggtg and ggctcaaagagcagatcaatg) and Linlab7981/Linlab4628 (aattgacccaccccactcaagttttagagctagaaatagcaagtt and cctctgacacatgcag ctcc). For sgRNA targeted mutation of *PSC1*, the primers Linlab8380/Linlab4627 (tagttgttttcgccga cgccaacagtataccctgccggtg and ggctcaaagagcagatcaatg) were used to amplify the *U6* promoter and Linlab8381/Linlab4628 (ggcgtcggcgaaaacaactagtttagagctagaaatagcaagtt and cctctgacacatgcagctcc

) to amplify the sgRNA scaffold. The *U6* promoter and sgRNA scaffold were fused together by overlap PCR with primers Linlab4594/Linlab4595 (ccatcgatttgcattagaactaaaaacaaagca and ccgctcgagtaaaaca aaaaagcaccgac) to generate the final sgRNA construct as described previously (*Fan and Lin, 2018*; *Lin et al., 2020*).

## Gene overexpression constructs

The *CBK1* overexpression construct was generated by amplifying the *CBK1* open reading frame with primers Linlab7005/BC (ataggccggccatgtcgtatcgcccaatccag) and Linlab7006/BC (cagcatctcgtatcgt cggaag) and cloning the fragment with FseI and PacI into the pXC plasmid backbone (*Wang et al., 2012*), which contains the promoter of *CTR4* and neomycin resistance marker. The *CTR4* promoter is highly induced on the copper limiting YPD media. The *MPK1* overexpression construct was generated by amplifying the *MPK1* open reading frame with primers Linlab8326/BC (ataggccggccatggacaataccc ctagacac) and Linlab8327/BC (ccttaattaaggctatgataatttctgcctctcc) and cloning the fragment with FseI and AsiSI into a pUC19 plasmid backbone, containing the promoter of *GPD1* and neomycin resistance marker. The *CDC24* overexpression construct was generated by amplifying the *CDC24* open reading frame with primers Linlab6674/BC (ataggccggccatgtctgtatccggtcccatctc) and Linlab6675/BC (ccttaatt aaggataaatctctccttgtggggtacc) and cloning the fragment with FseI and PacI into a pUC19 plasmid backbone, containing the promoter of *CTR4* and neomycin resistance marker. The overexpression constructs were integrated into the *SH2* locus as described previously (*Fan and Lin, 2018*; *Lin et al., 2020*).

## Transformation

Constructs for overexpression and deletion were transformed into *Cryptococcus* strains by the TRACE method (*Fan and Lin, 2018*; *Lin et al., 2020*), and transformants were selected on YPD medium with 100 µg/mL of NAT,100 µg/mL of neomycin (NEO), or 200 µg/mL of HYG.

## Quantitative real-time PCR

WT H99 strain along with the *cbk1Δ*, *CBK1*[OE] strain were cultured by shaking at 220 RPM at 30°C overnight in liquid YPD medium containing 50 µM $CuSO_4$ to suppress the CTR4 promoter of the *CBK1*[OE] construct. The cultures were then diluted to $OD_{600}$=0.2 in fresh liquid YPD medium containing 50 µM BCS(bathocuproine disulfonate) to induce expression. After 5 hr of further incubation, cells were collected, flash frozen in liquid nitrogen, and lyophilized overnight. Three biological replicates per strain were used. Total RNA was isolated by using the PureLink RNA Mini Kit (Invitrogen), and first strand cDNA was synthesized using the GoScript Reverse Transcription System (Promega) following the manufacturer's instructions. The Power SYBR Green system (Invitrogen) was used for RT-PCR. The following primers were used to target *CBK1*: Linlab9217/BC (gatgctctcactcctgattcc) and Linlab8641/BC (gtacgagtctgacttcaccga). The following primers were used to target the *TEF1* housekeeping gene as an endogenous control for each sample: Linlab329/XL (cgtcaccactgaagtcaagt) and Linlab330/XL (agaagcagcctccatagg). Relative transcript level was determined using the $\Delta\Delta$ct method as described previously. Statistical significance was determined using a Student's t-test.

## NanoString RNA profiling

Overnight YPD cultures of H99, *cbk1Δ*, *cbk1Δssd1Δ*, and *cbk1Δpsc1Δ* were washed 2× in PBS and resuspended in RPMI +165 mM MOPS, pH 7.4 before quantification on an Invitrogen Countess automated cell counter. Cells were diluted to $7.5×10^5$ cells per mL in 3 mL per well in a 6-well plate. Two wells were used for each biological replicate (n=3) and condition (ambient or 5% $CO_2$). Plates were sealed with BreatheEasy sealing membranes (Sigma #Z380059) and incubated in a static incubator at 30°C in ambient air or 5% $CO_2$ for 24 hr. Cells were harvested, pelleted at 3200×g for 5 min, and the supernatant was removed. The pellets were then frozen at –80°C and lyophilized overnight. Lyophilized cells were disrupted for 45 s with 0.5 mm glass beads on an MP Biomedicals FastPrep-24 benchtop homogenizer. RNA was extracted following manufacturer instructions for the Invitrogen PureLink RNA mini-kit with on-column DNAse treatment. Purified RNA was quantified on a NanoDrop OneC spectrophotometer, and a total of 100 ng per sample was combined with a custom probeset (*Source data 1*) from NanoString Technologies according to manufacturer instructions. Probes were hybridized at 65°C for 18 hr, then run on a NanoString nCounter SPRINT profiler according to

manufacturer instructions. Data from Reporter Code Count files were extracted with nSolver software (version 4.0), and raw counts were exported to Microsoft Excel. Internal negative controls were used to subtract background from raw counts (negative control average +2 SDs). Counts were normalized across samples by total RNA counts. Probes below background were set to a value of 1. Fold change and significance were calculated in Excel after averaging biological triplicates, using a Student t-test ($p < 0.05$). Volcano plot was generated with transformed values ($-\log[\text{p-value}]$ and $\log_2[\text{fold change}]$) in GraphPad Prism 9. Normalized total counts were used in Morpheus (https://software.broadinstitute.org/morpheus/) to generate a heat map, with hierarchical clustering, one minus Pearson correlation, average linkage method, and clustered according to rows and columns.

## Bioinformatics

Whole genome sequencing was performed using the Illumina platform with NovaSeq 6000 at the University of California – Davis Sequencing Center, Novogene USA. A paired-end library with approximately 350 base inserts was constructed for each sample, and all libraries were multiplexed and run in one lane using a read length of 150 bases from either side.

The Illumina reads were first trimmed with Trim Galore v0.6.5 (*Krueger, 2021*) and then mapped to the *C. neoformans* H99 reference genome (FungiDB version 50) using the BWA-MEM algorithm of the BWA aligner v0.7.17 (*Li, 2013*). SAMtools v1.10 (*Li et al., 2009a*), Picard Tools v2.16.0 (*Broad_Institute, 2022*), and bcftools v1.13 (*Danecek et al., 2021*) were used for variant calling from each sample. Variants in the suppressor strains were called with the original *cbk1Δ* mutant as a reference.

The protein diagrams of Psc1 and Ssd1 were made with the illustrator of biological sequences software package (*Liu et al., 2015*).

## Phagocytosis assays

The authenticated mouse macrophage cell line J774A.1 (ATCC TIB-67) was acquired from the American Type Culture Collection. Before being used, normal morphology, cell adhesion, and phagocytosis activity of the cell line was confirmed. Contamination by mycoplasma was not detected. Phagocytosis assays were performed using similar procedures as we described previously (*Lin et al., 2015*). Briefly, 1 mL of $2 \times 10^5$ J774A.1 macrophages (MΦ) in DMEM was seeded into a 24-well plate and incubated at 37°C with 5% $CO_2$ for 24 hr. *Cryptococcus* strains with a starting $OD_{600}$ of 0.2 in 3 mL of liquid YPD were cultured for 16 hr. Each strain had three technical replicates. The cells were washed three times in sterile $H_2O$. $2 \times 10^6$ cryptococcal cells of each strain were opsonized in either 40 µL of 100% fetal bovine serum, naïve mouse serum, or mouse serum from LW10 vaccinated A/J mice (*Lin et al., 2022*; *Zhai et al., 2015*), for 30 min prior to co-incubation with MΦ. Old DMEM from MΦ was removed, and 1 mL of fresh DMEM with the opsonized *Cryptococcus* cells was added, followed by a 2 hr incubation at 37°C with 5% $CO_2$. The co-culture was then washed six times with warm PBS to remove non-adherent *Cryptococcus* cells. To lyse the macrophages, the cell suspensions were washed with 1 mL of cold PBS +0.01% Triton X. Serial dilutions in PBS of the cell suspensions were then plated onto YNB agar medium and allowed to grow at 30°C for 2 days to count CFUs. Statistical analyses were performed using the program Graphpad Prism 8. A two-tailed t-test was applied to determine significance. A p-value of less than 0.05 was considered significant.

## *G. mellonella* infection model

*G. mellonella* larvae were purchased from Best Bait (Marblehead, OH, USA). The infection was performed as described previously described (*Mylonakis et al., 2005*). In brief, cryptococcal strains were inoculated in 3 mL of liquid YPD medium with the initial $OD_{600} = 0.2$ (approximately $10^6$ cell/mL) and incubated for 15 hr at 30°C with shaking. Prior to infection, cells were washed with sterile PBS three times and adjusted to the final concentration of $1 \times 10^7$ cell/mL. 5 µL of the cell suspension ($5 \times 10^4$ cells), or PBS for the control group, were injected into the last left proleg of the larvae. The proleg was cleaned with 70% ethanol prior to injection. Infected larvae were maintained at 30°C and monitored daily for survival.

Prior to fungal burden quantification, larvae were first cleaned with 70% ethanol. The larvae were cut open with sterile scissors and vortexed in a microcentrifuge tube containing 500 µL PBS and 100 µL of 0.5 mm diameter glass beads (RPI). Larval suspensions were then serially diluted in PBS and

plated onto YNB agar medium containing 50 µg/mL kanamycin and 20 µg/mL chloramphenicol and incubated at 30°C for 2 days before counting the CFUs.

Statistical analyses were performed using the program Graphpad Prism 8. The log-rank Mantel-Cox test was used to assess statistical significance of survival curves for comparison between two groups. One-way ANOVA tests were used to compare groups of three or more and for fungal burden assays.

## Murine models of cryptococcosis

### Intranasal infection model

Female Balb/C mice of 8–10 weeks old were purchased from the Jackson Labs (Bar Harbor, Maine). Cryptococcal strains were inoculated in 3 mL of liquid YPD medium with the initial $OD_{600}=0.2$ (approximately $10^6$ cell/mL) and incubated for 15 hr at 30 °C with shaking. Prior to intranasal infection, cells were washed with sterile saline three times and adjusted to the final concentration of $2\times10^5$ cell/mL. Once the mice were sedated with ketamine and xylazine via intraperitoneal injection, 50 µL of the cell suspension ($1\times10^4$ cells per mouse) were inoculated intranasally as previously described (*Lin et al., 2022*; *Zhai et al., 2012*; *Zhai et al., 2013*; *Zhao et al., 2020*; *Zhu et al., 2013*). Mice were monitored daily for disease progression. Surviving animals were euthanized at 35 or 60 DPI, and the brain, lungs, and kidneys, were dissected.

### Intravenous infection model

Prior to intravenous infections, cryptococcal cells were washed with sterile saline three times and adjusted to the final concentration of $1\times10^6$ cell/mL. Mice were sedated with Isoflurane. 100 µL of the cell suspension ($1\times10^5$ cells per mouse) were injected intravenously as previously described (*Zhai et al., 2012*; *Zhai et al., 2013*; *Zhao et al., 2020*; *Zhu et al., 2013*). After DPI 5, animals were euthanized, and the brain, lungs, and kidneys were dissected.

For fungal burden quantifications, dissected organs were homogenized in 2 mL of cold sterile PBS using an IKA-T18 homogenizer as we described previously (*Zhai et al., 2015*; *Zhai et al., 2012*). Tissue suspensions were serially diluted in PBS and plated onto YNB agar medium and incubated at 30°C for 2 days before counting the CFUs.

## Ethical statements

This study was performed according to the guidelines of NIH and the University of Georgia Institutional Animal Care and Use Committee (IACUC). The animal models and procedures used have been approved by the IACUC (AUP protocol numbers: A2017 08–023 and A2020 06–015).

## Acknowledgements

This work was supported by National Institutes of Health (http://www.niaid.nih.gov) (R01AI147541 to DJK and XL, and R01AI140719 to XL). The funder had no role in study design, data collection, and interpretation, or the decision to submit the work for publication. We thank all Lin lab members for their helpful suggestions. We thank Dr. Fanglin Zheng for the plasmid pFZ1, and Dr. Lukasz Kozubowski for the plasmid LKB61.

## Additional information

### Funding

| Funder | Grant reference number | Author |
| --- | --- | --- |
| National Institutes of Health | R01AI147541 | Damian J Krysan Xiaorong Lin |
| National Institutes of Health | R01AI140719 | Xiaorong Lin |

The funders had no role in study design, data collection and interpretation, or the decision to submit the work for publication.

## Author contributions

Benjamin J Chadwick, Conceptualization, Formal analysis, Validation, Investigation, Visualization, Methodology, Writing - original draft, Writing - review and editing; Tuyetnhu Pham, Laura C Ristow, Investigation, Visualization, Methodology; Xiaofeng Xie, Investigation, Methodology; Damian J Krysan, Resources, Funding acquisition, Project administration, Writing - review and editing; Xiaorong Lin, Conceptualization, Resources, Supervision, Funding acquisition, Investigation, Methodology, Project administration, Writing - review and editing

## Author ORCIDs

Benjamin J Chadwick http://orcid.org/0000-0002-8244-6190
Xiaorong Lin http://orcid.org/0000-0002-3390-8387

## Ethics

This study was performed according to the guidelines of NIH and the University of Georgia Institutional Animal Care and Use Committee (IACUC). The animal models and procedures used have been approved by the IACUC (AUP protocol numbers: A2017 08-023 and A2020 06-015).

## Decision letter and Author response

Decision letter https://doi.org/10.7554/eLife.82563.sa1
Author response https://doi.org/10.7554/eLife.82563.sa2

# Additional files

## Supplementary files

• Supplementary file 1. Hits from forward genetic screening.

• MDAR checklist

• Source data 1. NanoString probe targets.

## Data availability

Sequences generated from this research has been deposited to the Sequence Read Archive (SRA) under project accession number: PRJNA791949.

The following dataset was generated:

| Author(s) | Year | Dataset title | Dataset URL | Database and Identifier |
| --- | --- | --- | --- | --- |
| Lin X | 2022 | Whole genome sequencing of C. neoformans H99 suppressor strains of the RAM pathway downstream kinase Cbk1 knockout mutant | https://www.ncbi.nlm.nih.gov/bioproject/PRJNA791949 | NCBI BioProject, PRJNA791949 |

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

# Appendix 1

### Appendix 1—key resources table

| Reagent type (species) or resource | Designation | Source or reference | Identifiers | Additional information |
|---|---|---|---|---|
| Genetic reagent (*Cryptococcus neoformans KN99α, matα*) | WT strain: H99 | ***Nielsen et al., 2003*** | | |
| Genetic reagent (*C. neoformans KN99α, mat**a***) | WT strain: H99a | ***Nielsen et al., 2003*** | | |
| Genetic reagent (*C. neoformans KN99α, matα*) | *cbk1Δ* | ***Walton et al., 2006*** | FJW9 | H99alpha, *CBK1::NAT*$^r$ |
| Genetic reagent (*C. neoformans KN99α, matα*) | *mob2Δ* | ***Walton et al., 2006*** | FJW10 | H99alpha, *MOB2::NAT*$^r$ |
| Genetic reagent (*C. neoformans KN99α, matα*) | *kic1Δ* | ***Walton et al., 2006*** | FJW8 | H99alpha, *KIC1::NAT*$^r$ |
| Genetic reagent (*C. neoformans KN99α, matα*) | *tao3Δ* | ***Walton et al., 2006*** | AI136 | MATalpha, *TAO3:: NAT*$^r$ |
| Genetic reagent (*C. neoformans KN99α, matα*) | *sog2Δ* | ***Walton et al., 2006*** | AI131 | MATalpha, *SOG2:: NAT*$^r$ |
| Genetic Reagent (*Candida albicans, SN250*) | SN250 | ***Wakade et al., 2020*** | SN250 | HIS-, LEU-, ARG- |
| Genetic Reagent (*C. albicans, SN250*) | *cbk1ΔΔ* | ***Wakade et al., 2020*** | ΔΔcbk1 | SN250, ΔΔcbk1: *HIS-, LEU-, ARG-* |
| Genetic reagent (*C. neoformans KN99α, matα*) | *cna1Δ* | FGSC deletion set Plate 46 Well E12 | *cna1Δ* | H99alpha *CNA1::NAT*$^r$ |
| Genetic reagent (*C. neoformans KN99α, matα*) | *cdc24Δ* | FGSC deletion set Plate 33 Well C4 | *cdc24Δ* | H99alpha, *CDC24::NAT*$^r$ |
| Genetic reagent (*C. neoformans KN99α, matα*) | *mpk1Δ* | FGSC deletion set Plate 11 Well A5 | *mpk1Δ* | H99alpha, *MPK1::NAT*$^r$ |
| Genetic reagent (*C. neoformans KN99α, matα*) | *CBK1*$^{OE}$ | This study. | BC1449 | H99alpha, P$_{CTR4}$-*CBK1*-mCherry- *NEO*$^r$ |
| Genetic reagent (*C. neoformans KN99α, matα*) | *cdc24Δ, CBK1*$^{OE}$ | This study. | BC1281 | H99alpha, *CDC24::NAT*$^r$, P$_{CTR4}$-*CBK1*-mCherry- *NEO*$^r$ |
| Genetic reagent (*C. neoformans KN99α, matα*) | *cdc24Δ, CBK1*$^{OE}$ | This study. | BC1282 | H99alpha, *CDC24::NAT*$^r$, P$_{CTR4}$-*CBK1*-mCherry- *NEO*$^r$ |
| Genetic reagent (*C. neoformans KN99α, matα*) | *cna1Δ, CBK1*$^{OE}$ | This study. | BC1283 | H99alpha, *CNA1::NAT*$^r$, P$_{CTR4}$-*CBK1*-mCherry- *NEO*$^r$ |
| Genetic reagent (*C. neoformans KN99α, matα*) | *cna1Δ, CBK1*$^{OE}$ | This study. | BC1284 | H99alpha, *CNA1::NAT*$^r$, P$_{CTR4}$-*CBK1*-mCherry- *NEO*$^r$ |
| Genetic reagent (*C. neoformans KN99α, matα*) | *mpk1Δ, CBK1*$^{OE}$ | This study. | BC1285 | H99alpha, *MPK1::NAT*$^r$, P$_{CTR4}$-*CBK1*-mCherry-*NEO*$^r$ |

*Appendix 1 Continued on next page*

Appendix 1 Continued

| Reagent type (species) or resource | Designation | Source or reference | Identifiers | Additional information |
|---|---|---|---|---|
| Genetic reagent (*C. neoformans* KN99α, *mat*α) | *mpk1Δ, CBK1*$^{OE}$ | This study. | BC1286 | H99alpha, *MPK1::NAT*$^r$, P$_{CTR4}$-*CBK1*-mCherry-*NEO*$^r$ |
| Genetic reagent (*C. neoformans* KN99α, *mat*α) | *cdc24Δ, CDC24*$^{OE}$ | This study. | BC650 | H99alpha, *CDC24::NAT*$^r$, P$_{CTR4}$-mNeonGreen-*CDC24*-*NEO*$^r$ |
| Genetic reagent (*C. neoformans* KN99α, *mat*α) | *cbk1Δ, CBK1*$^{OE}$ | This study. | BC669 | H99alpha, *CBK1::NAT*$^r$, P$_{CTR4}$-*CBK1*-mCherry-*NEO*$^r$ |
| Genetic reagent (*C. neoformans* KN99α, *mat*α) | *mpk1Δ, MPK1*$^{OE}$ | This study. | BC1356 | H99alpha, *MPK1::NAT*$^r$, P$_{GPD1}$-mNeonGreen-*MPK1*-*NEO*$^r$ |
| Genetic reagent (*C. neoformans* KN99α, *mat*α) | *cna1Δ, CNA1*$^{OE}$ | **Kozubowski et al., 2011** | LK214 | H99a, *CNA1::NEO*$^r$, P$_{H3}$-GFP-*CNA1*-*NAT*$^r$ |
| Genetic reagent (*C. neoformans* KN99α, *mat*α) | *cbk1Δ, CDC24*$^{OE}$ | This study. | BC1357 | H99alpha, *CBK1::NAT*$^r$, P$_{CTR4}$-mNeonGreen-*CDC24*-*NEO*$^r$ |
| Genetic reagent (*C. neoformans* KN99α, *mat*α) | *cbk1Δ, MPK1*$^{OE}$ | This study. | BC1358 | H99alpha, *CBK1::NAT*$^r$, P$_{GPD1}$-mNeonGreen-*MPK1*-*NEO*$^r$ |
| Genetic reagent (*C. neoformans* KN99α, *mat*α) | *cbk1Δ, CNA1*$^{OE}$ | This study. | BC1359 | H99alpha, *CBK1::NAT*$^r$, P$_{H3}$-GFP-*CNA1*-*NAT*$^r$ |
| Genetic reagent (*C. neoformans* KN99α, *mat*α) | *sup1* | This study. | BC1068 | H99alpha, *CBK1::NAT*$^r$,SUP1 |
| Genetic reagent (*C. neoformans* KN99α, *mat*α) | *sup2* | This study. | BC1076 | H99alpha, *CBK1::NAT*$^r$,SUP2 |
| Genetic reagent (*C. neoformans* KN99α, *mat*α) | *cbk1Δssd1Δ* | This study. | BC1239 | H99alpha, *CBK1::NAT*$^r$, *SSD1::NEO*$^r$ |
| Genetic reagent (*C. neoformans* KN99α, *mat*α) | *ssd1Δ* | This study. | BC1241 | H99alpha, *SSD1::NEO*$^r$ |
| Genetic reagent (*C. neoformans* KN99α, *mat*α) | *cbk1Δpsc1Δ* | This study. | BC1369 | H99alpha, *CBK1::NAT*$^r$, *PSC1::HYG*$^r$ |
| Genetic reagent (*C. neoformans* KN99α, *mat*α) | *psc1Δ* | This study. | BC1393 | H99alpha, *PSC1::HYG*$^r$ |
| Genetic reagent (*C. neoformans* KN99α, *mat*α) | *cac1Δ* | **Bahn et al., 2006** | YSB42 | H99alpha, *CAC1::NAT*$^r$ STM#159 |
| Genetic reagent (*C. neoformans* KN99α, *mat*α) | *aca1Δ* | FGSC deletion set Plate 32 Well H6 | *aca1Δ* | H99alpha, *ACA1::NAT*$^r$ |
| Genetic reagent (*C. neoformans* KN99α, *mat*α) | *pkr1Δ* | FGSC deletion set Plate 33 Well H7 | *pkr1Δ* | H99alpha, *PKR1::NAT*$^r$ |
| Genetic reagent (*C. neoformans* KN99α, *mat*α) | *pde1Δ* | FGSC deletion set Plate 10 Well A12 | *pde1Δ* | H99alpha, *PDR1::NAT*$^r$ |

*Appendix 1 Continued on next page*

*Appendix 1 Continued*

| Reagent type (species) or resource | Designation | Source or reference | Identifiers | Additional information |
|---|---|---|---|---|
| Genetic reagent (*C. neoformans KN99α, matα*) | *pde2Δ* | FGSC deletion set Plate 11 Well H7 | *pde2Δ* | H99alpha, *PDR2::NAT*[r] |
| Genetic reagent (*C. neoformans KN99α, matα*) | *gpa1Δ* | **Bahn et al., 2005** | YSB83 | H99alpha, *GPA1::NAT*[r] STM#5 |
| Genetic reagent (*C. neoformans KN99α, matα*) | *cln1Δ* | This study. | BZ36 | H99alpha, *CLN1::NAT*[r] |
| Genetic reagent (*C. neoformans KN99α, matα*) | *ecm2201Δ* | FGSC deletion set Plate 2 Well A10 | *ecm2201Δ* | H99alpha, *ECM2201::NAT*[r] |
| Genetic reagent (*C. neoformans KN99α, matα*) | *kin4Δ* | Bahn Kinase Deletion Set Plate 4 Well G8 (**Lee et al., 2016**) | *kin4Δ* | H99alpha, *KIN4::NAT*[r] |
| Recombinant DNA reagent | pPZP-NATcc | **Dickinson et al., 2013** | pPZP-NATcc | |
| Recombinant DNA reagent | pDD162 | **Walton et al., 2006** | pDD162 | |
| Recombinant DNA reagent | pFZ1-*CDC24* | This study. | $P_{CTR4-2}$-mNeonGreen-*CDC24*(H99)-*NEO*[r] | |
| Recombinant DNA reagent | pXC-*CBK1*-mCh | This study. | $P_{CTR4-2}$-*CDC24*(H99)-mCherry-*NEO*[r] | |
| Recombinant DNA reagent | LKB61 | **Kozubowski et al., 2011** | $P_{GPD1}$-mCherry-*CNA1-HYG*[r] | |
| Recombinant DNA reagent | pUC19-*MPK1*-mNG | This study. | $P_{GPD1}$-*MPK1*-mNeonGreen-NEO[r] | |
| Cell line (*Mus. Musculus*, macrophage cell line J774A.1) | J774A.1 | American Type Culture Collection | ATCC TIB-67 | |
| Chemical Compound and drug | Hygromycin | Research Products International | Cat. NO.: H75000 | |
| Chemical Compound and drug | G418 | Research Products International | Cat. NO.: G64000 | |
| Chemical Compound and drug | Nourseothricin | Jena Bioscience | Cat. NO.: AB-102–25 G | |
| Software and algorithm | Graphpad Prism 9 | Graphpad | | |
| Software and algorithm | nSolver software version 4.0 | NanoString Technologies | | |
| Software and algorithm | Trim Galore v0.6.5 | **Krueger, 2021** | | |
| Software and algorithm | BWA aligner v0.7.17 | **Li, 2013** | | |
| Software and algorithm | SAMtools v1.10 | **Li et al., 2009a** | | |
| Software and algorithm | Picard Tools v2.16.0 | **Broad_Institute, 2022** | | |
| Software and algorithm | bcftools v1.13 | **Danecek et al., 2021** | | |
| Software and algorithm | Illustrator of biological sequences (IBS) | **Liu et al., 2015** | | |

