## [Editor Report]

This paper reports the identification of molecular determinants of CO2 tolerance in the human fungal pathogen Cryptococcus neoformans. The results are important for our understanding of how the fungus adapts from the ambient atmosphere to the CO2-enriched environment in the human host, and the findings are convincing and rely on biochemical, molecular, and genetic techniques. The results should be of interest to a broad community in the life sciences including microbiologists and infectious diseases investigators.

---

## [Decision Letter]

**Decision letter after peer review:**

Thank you for submitting your article "The RAM signaling pathway links morphology, thermotolerance, and CO2 tolerance in the global fungal pathogen Cryptococcus neoformans" for consideration by *eLife*. Your article has been reviewed by 3 peer reviewers, and the evaluation has been overseen by a Reviewing Editor and Arturo Casadevall as the Senior Editor. The following individual involved in review of your submission has agreed to reveal their identity: Ella Jacobs (Reviewer #3).

Essential revisions:

1) Need to address the comments from all three reviewers on clarifications on the text as indicated in their recommendations to authors

2) Need to address concerns about strain The cbk1Δ strain raised by reviewer 2.

3) Consider reviewer 3 recommendation to add an additional experiment testing survival after H2O2 stress.

*Reviewer #1 (Recommendations for the authors):*

Some points for consideration are as follows.

Lines 20-21: does it matter in the abstract that no ACE2 homology is known? The logical discovery from the suppressor screen would be its equivalent but that did not happen. Therefore recommend deleting those two sentences.

Lines 43-44: could delete "Krysan and Lin laboratories demonstrated that." Also on line 44, to "in the host vs. ~00.04% in".

Carbonic anhydrase has been studied in C. neoformans, so there is some research on the role of CO2 on this fungus. This might influence the text on lines 345-346 about bicarbonate sensing.

General: use a consistent style for pH; probably pH # (with the space) is the better option than pH#. One example of the mix in styles is seen in the labels on figure 1.

Lines 85-86: what was the rationale, other than convenience, of using a "simplified medium" [although note YPD is a complex undefined medium].

Line 133: as commented in the "Public Review" section, details are missing here about gene choice. For example, what are the "results from a separate study"?

Figure 2A: not sure this panel is needed.

Lines 169-170 and 343-355: are likely an extrapolation. It is worth considering two aspects of C. neoformans biology. (a) H99 is non-representative of serotype A, featuring a chromosomal rearrangement and gene loss. (b) A RAM pathway mutant in a serotype D background does not show strong temperature sensitivity (see Magditch et al. PLoS Pathogens 2012 figure 1). It may be wise to explore the CO2 impact in other background as well before drawing too many conclusions about differences between fungal phyla. Likewise, the CO2 phenotypes seen here for the cdc24 mutant are stronger than those observed by Chang et al. PLoS Genetics 2014. What is the consequence on O2 levels of increasing CO2 levels, e.g. do they reduce leading towards hypoxia?

Suppressor phenotypes and figure 4: Magditch et al. 2012 screened for suppressors using the sensitivity of RAM pathway mutants to calcineurin inhibition. Do the PSC1 and SSD1 suppressors grow in the presence of FK506? Likewise, do they restore the mating defects seen in RAM pathway mutants?

Line 270: perhaps change "all zeros in" to "no CFUs isolated from".

Line 278: "to have a poor phagocytosis".

Line 281: it would be good to expand on the rationale for using the vaccinated mouse material, and what the vaccine was.

Line 298: "detected a few fungal".

Lines 471, 477: delete "media" [the last M in "DMEM"].

Line 475: micro symbol for "ul".

Reference list: check the formatting is consistent, e.g. subscript 2 in CO2, italics on species names, in case the journal does not do this.

Figure S3 legend: italics on "cbk1".

Table S1 could be more informative, e.g. one more column to highlight possible functions or links. For example, it was not immediately clear which were the RAM pathway mutants or their gene names. How many of these mutants are also temperature sensitive or have other phenotypes? Do these CNAG_##### genes have other names from previous studies that could orient readers better?

*Reviewer #2 (Recommendations for the authors):*

1. The authors have screened for suppressor mutants of cbk1Δ using the assay described in the text (lines 183-188). A total of 11 suppressor colonies were further examined and the genomes were sequenced. Based on the sequencing result, the suppressor colonies were categorized into two groups: two colonies in the sup1 group, both of which contained an in-frame deletion in the PSC1 gene, and 9 colonies in the sup2 group that contain mutations (any of the missense, frame-shift, or premature STOP) in the SSD1 ORF. The authors have performed in vitro as well in vivo experiments in a mice model with a single suppressor colony from each group. While the nature of mutations varies in sup2 colonies, the authors have shown that the cbk1Δssd1Δ strain behaves similarly to the sup2 group. While it is unclear from the text the nature of the mutation in the sup2 colony, the authors took forward the mutants for subsequent experiments (whether missense, frameshift, or premature STOP). It may be assumed that a missense mutation found in sup2 and a frameshift mutation found in sup1, which the authors reported, lead to a suppressor phenotype similar to respective gene deletion in the cbk1Δ background. It is suggestive of the critical role(s) of the individual residues in both the proteins, mutations of which lead to complete loss of activity. We recommend the authors generate the respective point mutants in those two proteins and recapitulate the sup1 and sup2 phenotypes.

2. It is understood from the in vitro results that:

a. The cbk1Δ strain is nearly inviable (Figure 4C) at 37oC irrespective of ambient or +5% CO2.

b. The sup1, cbk1Δsup1Δ exhibit partial rescue at 37oC alone but very poor rescue at 37oC+5% CO2

c. Similarly, the sup2, cbk1Δssd1Δ strains exhibit partial rescue, slightly better than sup1.

Thus, at 37oC, in presence of ambient or 5% CO2, those strains exhibit a significant loss of viability compared to H99. However, the authors used the same number of cells as inoculum of all strains for the mice experiment ignoring the difference in the viability of the above strains. Hence, the conclusion that the sup1 and sup2 strains fail to cause any mortality (lines 255-258) is questionable. We recommend that the authors normalize the sup1 and sup2 inoculum such that viable numbers, not the total numbers, of cells, are comparable to those in H99 at the host condition. Authors may consider using an alternate model system to validate those virulence traits by taking into account the above factors.

3. The authors may consider showing the overexpression of Cbk1 using Western blots (Figure 2).

4. Adding scale bars to Figure 4D will help in understanding the morphological changes better.

5. Line 93: Adding the information on the total number of ORFs in the C. neoformans genome will help the reader understand what percentage of the genome was represented by the deletion mutant collection.

*Reviewer #3 (Recommendations for the authors):*

This manuscript would be strengthened by an experiment comparing the suppressor strain's survival after H2O2 exposure or another general stress test to bolster the claim that sup2's increased growth at host CO2 levels at pH 6 is a solid explanation for the differences in establishment and persistence of infection in the absence of mouse mortality.

The acknowledgement that other virulence factors are also impacted by this kinase should be made more apparent in the text and/or included in a main figure.

Additionally, in SI Appendix Figure 4, the slightly increased growth of sup2 during CO2 exposure is only seen at pH 6, while at pH 7.4 both strains appear similarly negatively impacted by CO2 with growth reduced to cbk1∆ levels. Further commentary on why the change in pH appears to greatly impact the sup1 and sup2 strains in pH 7.4 compared to pH 6 would be helpful.

---

## [Author Response]

Reviewer #1 (Recommendations for the authors):Some points for consideration are as follows.Lines 20-21: does it matter in the abstract that no ACE2 homology is known? The logical discovery from the suppressor screen would be its equivalent but that did not happen. Therefore recommend deleting those two sentences.

Although it will not matter to the design of our gene deletion screen, the suppressor screen, or the main findings of the study, we believe that exclusion of the statement about the lack of *ACE2* homolog in *Cryptococcus* will likely raise more questions in readers’ mind because Ace2 plays such an important role in the RAM pathway in the well‐studied *Saccharomyces* and *Candida* species. Furthermore, our findings indicate that in the absence of an Ace2 homolog in *C. neoformans*, the genes we identified here play a similar role in cell separation and morphology as Ace2 does in these ascomycete yeasts.

Lines 43-44: could delete "Krysan and Lin laboratories demonstrated that." Also on line 44, to "in the host vs. ~00.04% in".

Changed.

Carbonic anhydrase has been studied in C. neoformans, so there is some research on the role of CO2 on this fungus. This might influence the text on lines 345-346 about bicarbonate sensing.

We thank the reviewer for pointing this out. Yes, there are studies on genes important for growth in ambient air (low levels of CO_2_) but dispensable for growth in high concentrations of CO_2_. Carbonic anhydrase Can1 and Can2 of *C. neoformans* are such factors and therefore they are dispensable for cryptococcal virulence in the host due to restored growth by high concentrations of CO_2_. Our study here focuses on genes important for tolerance of CO_2_ at high concentrations. That said, we agree with the reviewer that it is on a related topic. We have included a reference to a study on *Cryptococcus* on carbonic anhydrases in the second Results section (lines 99‐102), and we have modified the Discussion section on lines 439‐443. “In *C. albicans*, CO_2_ levels are sensed through bicarbonate or cAMPdependent activation of adenylyl cyclase to increase hyphal growth (Du et al., 2012; Hall et al., 2010).

While these pathways may also be functioning to sense CO_2_ in *Cryptococcus (Bahn, Cox, Perfect, and Heitman, 2005; Mogensen et al., 2006)*, our results indicate that these pathways do not play a significant role in host CO_2_ tolerance in *C. neoformans*.”

General: use a consistent style for pH; probably pH # (with the space) is the better option than pH#. One example of the mix in styles is seen in the labels on figure 1.

We thank the reviewer for finding this inconsistency. We have changed the style of all to “pH #”.

Lines 85-86: what was the rationale, other than convenience, of using a "simplified medium" [although note YPD is a complex undefined medium].

We modified lines 84‐85 to “For large‐scale screening, we used the nutrient rich YPD medium on which *C. neoformans* grows well”. Besides convenience, the pH and nutrient content of YPD is also beneficial for *Cryptococcus* growth and therefore limits any potential detrimental effects that other media might have caused.

Line 133: as commented in the "Public Review" section, details are missing here about gene choice. For example, what are the "results from a separate study"?

We have modified this section to make the details of the separate transcriptomics study more clear on lines 133‐135. “Transcript levels of 118 genes were measured and those genes were chosen based on RNA sequencing results of four different natural isolates in a separate study (Krysan et al., in preparation). In that study, these genes were differentially expressed in CO_2_ vs ambient air conditions in either two CO_2_‐sensitive or two CO_2_‐tolerant natural strains (Source data 1). The list of the 118 probes is provided in this manuscript as Source data 1.

Figure 2A: not sure this panel is needed.

Figure 2A is included to show that the previously identified CO_2_‐sensitive strains, based on growth on the nutrient‐limiting mammalian cell culture RPMI media (Krysan et al., 2019), are also sensitive on rich YPD media that is favorable for fungal growth. The growth defects become more severe when the CO_2_ concentration is increased to 20%, which is the concentration used in our deletion library screen. The CO_2_ effect on cryptococcal growth on YPD media has not been reported previously and we believe that it is important to include the result in this study.

Lines 169-170 and 343-355: are likely an extrapolation. It is worth considering two aspects of C. neoformans biology. (a) H99 is non-representative of serotype A, featuring a chromosomal rearrangement and gene loss. (b) A RAM pathway mutant in a serotype D background does not show strong temperature sensitivity (see Magditch et al. PLoS Pathogens 2012 figure 1). It may be wise to explore the CO2 impact in other background as well before drawing too many conclusions about differences between fungal phyla. Likewise, the CO2 phenotypes seen here for the cdc24 mutant are stronger than those observed by Chang et al. PLoS Genetics 2014. What is the consequence on O2 levels of increasing CO2 levels, e.g. do they reduce leading towards hypoxia?

We thank the reviewer for pointing out these observations to us. Based on the results of Magditch et al. (Magditch et al., 2012), the RAM pathway mutant phenotype in different strain backgrounds of *Cryptococcus* is similar to our mutants in the H99 background in terms of morphology and temperature sensitivity (their Figure 1 vs. our Figure 3). However, the RAM mutant phenotype is distinct from ascomycete yeasts such as *Saccharomyces cerevisiae* and *Candida albicans* (the later shown in Figure 3‐figure supplement 1), which is not temperature sensitive and exhibits depolarized growth as round yeast cells.

Our CO_2_ incubator controls CO_2_ directly but not the O_2_ levels. It is possible that increasing CO_2_ from 0.04% to 5% will accordingly lower other gases in the air including O_2_ by 95%, which means that the O_2_ levels should be around 21%*0.95 = 19.95%. This is hardly considered a hypoxic condition for *Cryptococcus*. Thus, our gas conditions should be similar to their 20%O_2_5%CO_2_. However, we took pictures of our colonies after 2 days of incubation whereas they took pictures after 3 days of incubation. The prolonged incubation in their setting might have diminished the differences in growth that could be observed at earlier time points. This and other possible differences in incubators, such as humidity, may also contribute to the variation in phenotype of our *cdc24*Δ mutant here vs. in the publication by Chang et al. PLoS Genetics 2014. That said, there is a discernable growth defect of the *cdc24*Δ mutant in 20%O_2_5%CO_2_ based on their Figure 1A, which is consistent with our observation.

Suppressor phenotypes and figure 4: Magditch et al. 2012 screened for suppressors using the sensitivity of RAM pathway mutants to calcineurin inhibition. Do the PSC1 and SSD1 suppressors grow in the presence of FK506? Likewise, do they restore the mating defects seen in RAM pathway mutants?

We thank the reviewer for bringing this to our attention. We have now tested these two phenotypes and added them to the Figure 4‐figure supplement 1. The suppressor mutants are unable to rescue FK506 growth inhibition or the mating defects of the *cbk1*Δ mutant. However, *psc1* and *ssd1* suppressors restored many of the *cbk1*Δ mutant defects, including growth on Congo red and H2O2, in addition to thermotolerance and CO_2_ tolerance. The growth rescue is consistent with the selective condition used to isolate these suppressor mutants.

Line 270: perhaps change "all zeros in" to "no CFUs isolated from".

Changed.

Line 278: "to have a poor phagocytosis".

Changed.

Line 281: it would be good to expand on the rationale for using the vaccinated mouse material, and what the vaccine was.

We have modified lines 344‐348 to describe the rationale: “Because different types of opsonization can impact phagocytosis of *C. neoformans*, we decided to opsonize the fungus using either naïve mouse serum (complement mediated phagocytosis) or serum from mice vaccinated against cryptococcosis (complement + antibody mediated phagocytosis). The serum (containing antibodies) from the vaccinated mice recognizes antigens present in the capsule of cryptococcal cells (Lin et al., 2022; Zhai et al., 2015).”

Line 298: "detected a few fungal".

Changed.

Lines 471, 477: delete "media" [the last M in "DMEM"].

Changed.

Line 475: micro symbol for "ul".

Changed.

Reference list: check the formatting is consistent, e.g. subscript 2 in CO2, italics on species names, in case the journal does not do this.

Changed.

Figure S3 legend: italics on "cbk1".

Changed.

Table S1 could be more informative, e.g. one more column to highlight possible functions or links. For example, it was not immediately clear which were the RAM pathway mutants or their gene names. How many of these mutants are also temperature sensitive or have other phenotypes? Do these CNAG_##### genes have other names from previous studies that could orient readers better?

We thank the reviewer for this suggestion. We have added the gene name from the *Saccharomyces* genome database (SGD) for each gene ID which has a homolog in *S. cerevisiae* to Supplementary file 1 (formerly Table S1). Fortunately, many of the genes discussed in the manuscript have homologues in *Saccharomyces cerevisiae*, including the RAM pathway gene names.

Reviewer #2 (Recommendations for the authors):1. The authors have screened for suppressor mutants of cbk1Δ using the assay described in the text (lines 183-188). A total of 11 suppressor colonies were further examined and the genomes were sequenced. Based on the sequencing result, the suppressor colonies were categorized into two groups: two colonies in the sup1 group, both of which contained an in-frame deletion in the PSC1 gene, and 9 colonies in the sup2 group that contain mutations (any of the missense, frame-shift, or premature STOP) in the SSD1 ORF. The authors have performed in vitro as well in vivo experiments in a mice model with a single suppressor colony from each group. While the nature of mutations varies in sup2 colonies, the authors have shown that the cbk1Δssd1Δ strain behaves similarly to the sup2 group. While it is unclear from the text the nature of the mutation in the sup2 colony, the authors took forward the mutants for subsequent experiments (whether missense, frameshift, or premature STOP). It may be assumed that a missense mutation found in sup2 and a frameshift mutation found in sup1, which the authors reported, lead to a suppressor phenotype similar to respective gene deletion in the cbk1Δ background. It is suggestive of the critical role(s) of the individual residues in both the proteins, mutations of which lead to complete loss of activity. We recommend the authors generate the respective point mutants in those two proteins and recapitulate the sup1 and sup2 phenotypes.

The *sup1* group contained amino acid deletion mutations in the PARN domain of Psc1, and the *sup2* group contained frameshifts or gain of stop codons in Ssd1. In one suppressor mutant that contained a frameshift mutation in Ssd1, it also contains a missense mutation in Ssd1. We apologize that we mistakenly left out the asterisk that was referenced in the Figure 4 legend in the original manuscript. We have added an asterisk in Figure 4 near “MS” and a description in the figure legend in this revised version to show that this mutation was found together with a frameshift mutation in the same suppressor mutant. Based on the evidence we have gathered, we conclude that the suppressor mutants were generated by loss‐of‐function mutations in Ssd1 or Psc1 for the following reasons: (1) Based on our DNA sequencing results, we only identified high impact mutations in either Psc1 or Ssd1 in the suppressor mutants; (2) Each suppressor mutant contained a high impact polymorphism or indel in Psc1 or Ssd1; (3) The double deletion mutants recapitulate the phenotypes of the natural suppressor mutants.

2. It is understood from the in vitro results that:a. The cbk1Δ strain is nearly inviable (Figure 4C) at 37oC irrespective of ambient or +5% CO2.b. The sup1, cbk1Δsup1Δ exhibit partial rescue at 37oC alone but very poor rescue at 37oC+5% CO2c. Similarly, the sup2, cbk1Δssd1Δ strains exhibit partial rescue, slightly better than sup1.Thus, at 37oC, in presence of ambient or 5% CO2, those strains exhibit a significant loss of viability compared to H99. However, the authors used the same number of cells as inoculum of all strains for the mice experiment ignoring the difference in the viability of the above strains. Hence, the conclusion that the sup1 and sup2 strains fail to cause any mortality (lines 255-258) is questionable. We recommend that the authors normalize the sup1 and sup2 inoculum such that viable numbers, not the total numbers, of cells, are comparable to those in H99 at the host condition. Authors may consider using an alternate model system to validate those virulence traits by taking into account the above factors.

We thank the reviewer for bringing up the complexity of virulence assessment in mammalian animal models. All the cells for virulence assays were cultured in vitro at 30^o^C and not at 37^o^C. Furthermore, the viable CFUs used in inoculum were confirmed. That said, once inoculated into mice, fungal cells have to live at the host body temperature and the temperature‐sensitive mutants will fare worse in the host compared to the wild type control. Unfortunately, there is no good way to determine an appropriate inoculum for each strain for comparison of multiple strains in animal models because many different factors work together to determine the outcome of infections and temperature is just one of the factors. Furthermore, depending on the time of examination, different number of viable cells of the same strain will be recovered from the animals. To compound the issue further, variation in the number of fungal cells used in inoculation could cause different host immune responses even if the cells are heat‐killed prior to inoculation. Therefore, there has been no better way to compare virulence of different strains in mice than the current standard of using the same inoculum across different strains.

To remove the temperature sensitivity variable in a virulence assay, we have added experiments with the *Galleria mellonella* larva infection model. This model is commonly used to assay virulence at lower temperatures as the body temperature of *G. mellonella* is the same as the incubation environment. In this *G. mellonella* infection model conducted at 30°C, we observed similar virulence pattern among these strains compared to that in the mouse infection model. This is now described in the Results section 5 and included in Figure 6.

3. The authors may consider showing the overexpression of Cbk1 using Western blots (Figure 2).

Our overexpression construct was integrated into the safe haven “*SH2”* region for all mutants to minimize any potential positional effect in our comparison. Our overexpression of *CBK1* is able to complement the *cbk1*Δ mutant phenotypes (Figure 2B), confirming its expression and functionality. Unfortunately, we do not have an antibody against Cbk1 for western blot analysis. Instead, we carried out a real‐time PCR experiment and confirmed that *CBK1* is indeed overexpressed. The result is now included in the Figure 2‐figure supplement 2.

4. Adding scale bars to Figure 4D will help in understanding the morphological changes better.

We thank the reviewer for pointing this out. We have added a scalebar to this figure.

5. Line 93: Adding the information on the total number of ORFs in the C. neoformans genome will help the reader understand what percentage of the genome was represented by the deletion mutant collection.

We have added the total number of protein coding genes in the *C. neoformans* H99 genome on line 92.

Reviewer #3 (Recommendations for the authors):This manuscript would be strengthened by an experiment comparing the suppressor strain's survival after H2O2 exposure or another general stress test to bolster the claim that sup2's increased growth at host CO2 levels at pH 6 is a solid explanation for the differences in establishment and persistence of infection in the absence of mouse mortality.

We agree with this reviewer that testing additional stresses could help differentiate *sup1* and *sup2* phenotypes. Other reviewers also suggested testing different phenotypes of the suppressor mutants. We have added these additional phenotypical assays along with an H2O2 stress spotting assay in the Figure 4‐figure supplement 1. As mentioned earlier, although the differences between the suppressor mutants, the *cbk1∆* mutant, and the wildtype strain are dramatic in these stress assays, there is no obvious difference in growth in these phenotypical assays between *sup1* and *sup2* except thermotolerance and CO_2_ tolerance.

The acknowledgement that other virulence factors are also impacted by this kinase should be made more apparent in the text and/or included in a main figure.

We have added to our section 3 results “In *C. neoformans*, various virulence factors are impacted by deletion of *CBK1*, including urease activity and thermotolerance (K. T. Lee et al., 2016)” (lines 177-179). We have also changed our wording in the manuscript to make it clear that better tolerance of CO_2_ contributes to better survival of the *sup2* mutant is only our hypothesis and there could be other unrecognized contributing factors. “The only in vitro difference observed between *sup1* and *sup2* was better growth of *sup2* at host CO_2_ levels which may explain the difference in their ability to propagate and persist in the mouse lungs. However, it is worth nothing that due to the complex host environment, there could be other unrecognized factors contributing to the differences in vivo.” (Lines 330‐332).

Additionally, in SI Appendix Figure 4, the slightly increased growth of sup2 during CO2 exposure is only seen at pH 6, while at pH 7.4 both strains appear similarly negatively impacted by CO2 with growth reduced to cbk1∆ levels. Further commentary on why the change in pH appears to greatly impact the sup1 and sup2 strains in pH 7.4 compared to pH 6 would be helpful.

We have added an explanation for this result, “Both *sup1* and *sup2* showed no improved growth compared to the *cbk1∆* mutant at pH 7.4 37°C + 5% CO_2_. This is likely due to the detrimental combination of high temperature, CO_2_, and high pH, as the WT also showed significantly reduced growth in this condition. (Figure 4—figure supplement 1A).” (Lines 226-229)